# Coupling of melanocyte signaling and mechanics by caveolae is required for human skin pigmentation

Lia Domingues [1✉], Ilse Hurbain[1,2], Floriane Gilles-Marsens[1,5], Julia Sirés-Campos [1], Nathalie André[3], Melissa Dewulf[4], Maryse Romao[1,2], Christine Viaris de Lesegno[4], Anne-Sophie Macé[2], Cédric Blouin [4], Christelle Guéré[3], Katell Vié[3], Graça Raposo[1,2,6], Christophe Lamaze[4,6] & Cédric Delevoye [1,2✉]

Tissue homeostasis requires regulation of cell–cell communication, which relies on signaling molecules and cell contacts. In skin epidermis, keratinocytes secrete factors transduced by melanocytes into signaling cues promoting their pigmentation and dendrite outgrowth, while melanocytes transfer melanin pigments to keratinocytes to convey skin photoprotection. How epidermal cells integrate these functions remains poorly characterized. Here, we show that caveolae are asymmetrically distributed in melanocytes and particularly abundant at the melanocyte–keratinocyte interface in epidermis. Caveolae in melanocytes are modulated by ultraviolet radiations and keratinocytes-released factors, like miRNAs. Preventing caveolae formation in melanocytes increases melanin pigment synthesis through upregulation of cAMP signaling and decreases cell protrusions, cell–cell contacts, pigment transfer and epidermis pigmentation. Altogether, we identify that caveolae serve as molecular hubs that couple signaling outputs from keratinocytes to mechanical plasticity of pigment cells. The coordination of intercellular communication and contacts by caveolae is thus crucial to skin pigmentation and tissue homeostasis.

[1] Institut Curie, PSL Research University, CNRS, UMR144, Structure and Membrane Compartments, 75005 Paris, France. [2] Institut Curie, PSL Research University, CNRS, UMR144, Cell and Tissue Imaging Facility (PICT-IBiSA), 75005 Paris, France. [3] Laboratoire Clarins, 5 rue Ampère, 95000 Pontoise, France. [4] Institut Curie, PSL Research University, INSERM U1143, CNRS UMR 3666, Membrane Mechanics and Dynamics of Intracellular Signaling Laboratory, 75005 Paris, France. [5] Present address: Institut NeuroMyoGene, UCBL1, UMR 5310, INSERM U1217, Génétique et Neurobiologie de C. Elegans, Faculté de Médecine et de Pharmacie, 8 Avenue Rockefeller, 69008 Lyon, France. [6] These authors contributed equally: Graça Raposo, Christophe Lamaze. ✉email: ldddomingues@gmail.com; cedric.delevoye@curie.fr

Human skin comprises a highly stratified epidermis and a bottom dermis. The epidermis, the outermost and photo-protective layer of the skin, is mainly composed of melanocytes and keratinocytes that together create a structural and functional epidermal unit[1]. Melanocytes are neural crest-derived cells[2] that extend dendrites and allow them to contact up to 40 epidermal keratinocytes[1]. The main role of melanocytes is to produce the melanin pigments in a specialized organelle, called melanosome, that undergoes maturation from early non-pigmented to late pigmented stages[3,4]. The maturing and pigmented melanosome moves toward the tip of the dendrites[5] to be transferred to keratinocytes where it protects the nuclei against ultraviolet (UV) radiations. In melanocytes, the formation of dendrites, melanosome biogenesis, synthesis and transfer of melanin to keratinocytes is a tightly coordinated process under the control of UV radiations, keratinocytes-secreted factors (e.g. hormones) and secreted endosomal-derived vesicles called exo-somes[6–8]. These secreted hormones can trigger different trans-duction pathways in melanocytes, including the cyclic adenosine monophosphate (cAMP) signaling pathway through binding to various G-protein coupled receptors (GPCRs) at the cell surface[9,10]. As a result, melanocytes increase pigment synthesis—mainly through stimulating gene transcription of melanin-synthesizing enzymes[11]—and dendrite outgrowth—through regulation of Rho GTPases activity and remodeling of the actin cytoskeleton[12,13]. Also, we have recently shown that specific miRNAs associated with exosomes released from keratinocyte modulate human melanocyte pigmentation by enhancing the expression of proteins associated with melanosome maturation and trafficking[8]. However, how environmental cues are spatially and temporally controlled in melanocytes to be efficiently translated into biochemical and physical cellular responses remains mostly uncharacterized.

Caveolae are cup-shaped plasma membrane invaginations firstly described in endothelial and epithelial cells[14,15]. Their size (50–100 nm) and the absence of an electron-dense coat mor-phologically distinguish caveolae from other invaginated struc-tures at the plasma membrane[16]. Caveolae are mainly composed of two groups of proteins, the caveolins (Cav1, -2 and -3)[17] and the more recently identified cavins (Cavin1, 2, 3 and 4)[18]. Caveolae biogenesis and functions are dependent on Cav1 and Cavin1 in non-muscle cells, and on Cav3 in muscle cells[19]. Caveolae have various crucial functions including endocytosis, lipid homeostasis, signal transduction and, the most recently identified, mechanoprotection[19,20]. As a transduction platform, caveolae control the production of second messengers, such as cAMP, through local confinement of different elements of this signaling cascade[21]. Cav1 and -3 contain a scaffolding domain (CSD) located in the N-terminal region that could bind to transmembrane adenylate cyclases (tmACs) to decrease their activities, and thus to control intracellular cAMP levels[22]. For instance, caveolae in cardiomyocytes participate in the compart-mentalization of intracellular cAMP, which can regulate cell contractility in distal regions of the heart and, therefore, its function[23,24]. The mechanoprotective role of caveolae is asso-ciated with the maintenance of plasma membrane integrity when both cells and tissues experience chronical mechanical stress[25–28]. And caveolae were recently shown to couple mechanosensing with mechanosignaling in human muscle cells, a process that is impaired in caveolae-associated muscle dystrophies[29].

Epidermal melanocytes and keratinocytes are in constant communication, not only via secreted factors and exosomes that modulate cellular responses, but also by the physical contacts they establish to maintain the tissue homeostasis and pigmentation. Here, we report an additional function for caveolae, which by integrating the biochemical and mechanical behavior of melanocytes, control melanin transfer to keratinocytes and epi-dermis pigmentation. Altogether, this study provides the first evidence for a physiologic role of caveolae as a molecular sensing platform required for the homeostasis of one of the largest human tissue, the skin epidermis.

## Results

**Caveolae distribute asymmetrically in melanocytes and are positively regulated by keratinocyte-secreted factors.** Melano-cytes and keratinocytes establish a complex intercellular dialog required for skin photoprotection. 2D co-culture systems, where these two cell types share the same medium, have been widely used to study intercellular communication and pigment transfer between epidermal cells[7,30]. To evaluate the distribution of caveolae within the epidermal unit in 2D, normal human mela-nocytes and keratinocytes were co-cultured and labeled for two main constituents of caveolae, Cav1 or Cavin1. Immuno-fluorescence microscopy revealed that both Cav1 and Cavin1, and therefore caveolae, were asymmetrically distributed in melano-cytes (Fig. 1a, b; arrowheads) that were identified by the abundant HMB45 staining of the processed fragment of the premelanosome protein PMEL[4] (Methods). This asymmetry was not observed in keratinocytes.

Symmetry-breaking can occur in response to local external chemical and/or mechanical cues such as signaling molecules and/or cell–cell contacts, respectively[31]. However, in the absence of any type of spatial signaling, cells break their symmetry randomly and spontaneously[32]. When grown alone in the absence of any pre-existing signaling cues, one third of melanocytes presented an asymmetric distribution of caveolae, as shown by the endogenous staining of Cav1 and Cavin1 (Fig. 1c and Supplementary Fig. 1a; arrowheads in top panels). This asymmetry was restricted to caveolae as the distribution of clathrin-coated pits (CCPs, see AP-2 staining; Supplementary Fig. 1b), the canonical plasma membrane invaginated structures mediating endocytosis[33] was even. Interestingly, the number of melanocytes showing caveolae asymmetrically distributed doubled when co-cultured with keratinocytes (Fig. 1a, c), whereas co-culture with HeLa cells had no effect (Fig. 1c and Supplementary Fig. 1a; bottom panels). This shows that the intrinsic asymmetry of caveolae in melanocytes is specifically enhanced by keratinocytes, either by cell–cell contacts and/or by keratinocytes-secreted factors. To address the role of extracellular factors in caveolae distribution, melanocytes were incubated with the medium recovered from a confluent culture of keratinocytes (referred as keratinocytes-conditioned medium, Ker-CM). Under this condition, we observed two to fourfold increase of the number of melanocytes with an asymmetric distribution of caveolae (Fig. 1d and Supplementary Fig. 1c, arrowheads) as compared to cells grown in their own medium (Mel supplemen-ted medium) or in keratinocytes non-conditioned medium (Ker medium). The proportion of melanocytes with caveolae asymme-trically distributed was similar between cells co-cultured with keratinocytes (Fig. 1c) and cells incubated with Ker-CM (Fig. 1d), indicating that factors secreted from keratinocytes are the main extracellular contributors to the increased asymmetric distribu-tion of caveolae in melanocytes.

**Caveolae are preferentially located at the melanocyte–keratinocyte interface in human epidermis and are more abundant in melanocytes during tissue pigmentation.** We investigated the distribution of caveolae at the melanocyte–keratinocyte interface in human skin samples. The tissues were chemically fixed or physically immobilized using high-pressure freezing (HPF), which preserves membranes in

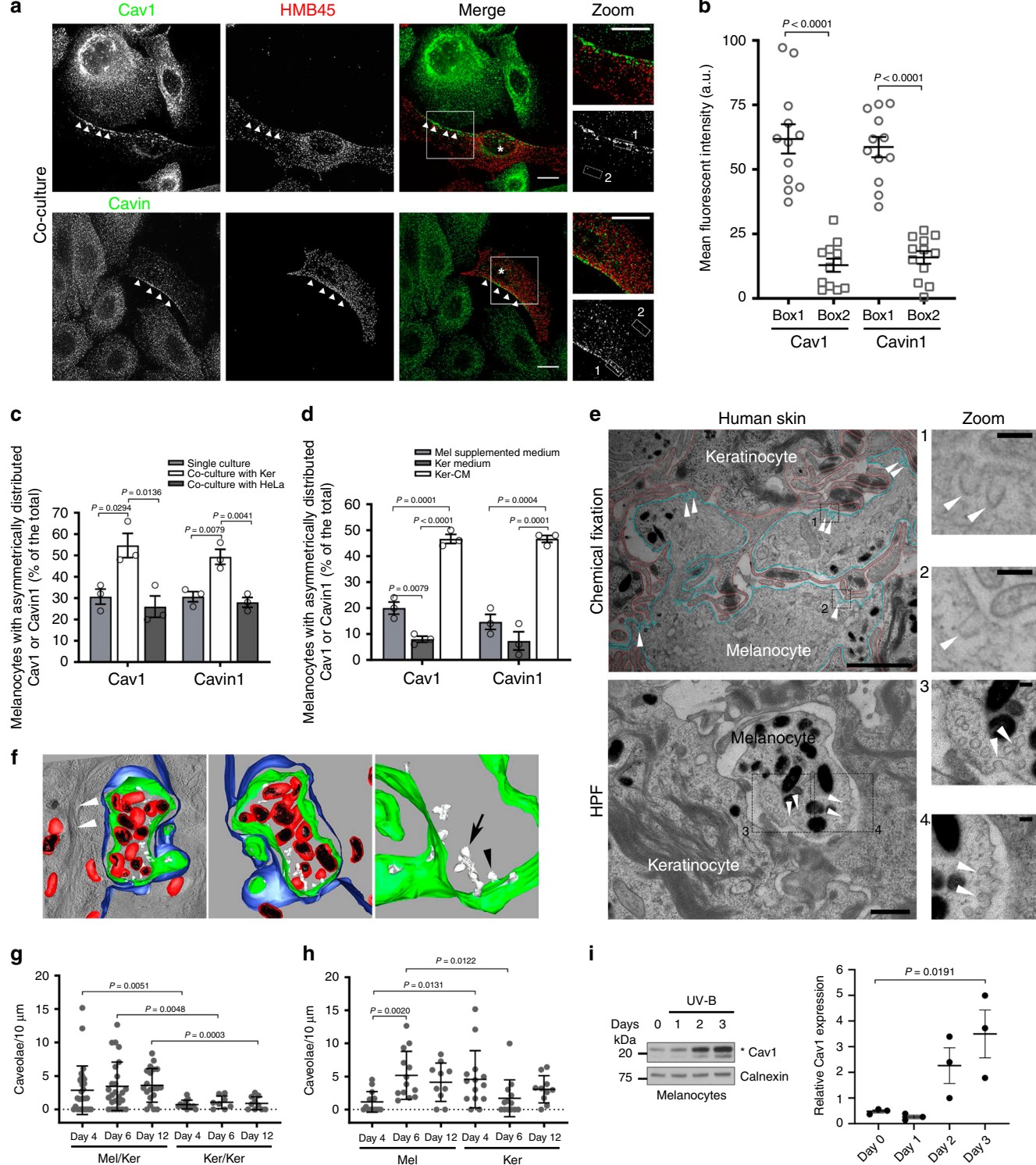

their native state[34], processed for ultrathin (60 nm) sectioning and analyzed by 2D conventional transmission electron microscopy (TEM) (Fig. 1e and Supplementary Fig. 1d). The melanocyte–keratinocyte interface revealed numerous plasma membrane-associated cup-shaped invaginations, with a diameter between 43 and 102 nm and an average size of 63.9 nm, which lacked an electron-dense cytoplasmic coat (Fig. 1e and Supplementary Fig. 1d, arrowheads). Immunogold labeling on ultrathin cryosections of human skin samples revealed that these invaginations were positive for Cav1 in melanocytes (Supplementary

Fig. 1e) and were thus identified as caveolae. To access caveolae 3D ultrastructure, thick-sectioned (300 nm) human skin samples were subjected to double-tilt electron tomography (Fig. 1f and Supplementary Fig. 1f). The reconstructed 3D model (Fig. 1f and Supplementary Video 1) depicts an epidermal area consisting of a transversal section of a melanocyte dendrite (plasma membrane in green) containing pigmented melanosomes (limiting membrane in red and intraluminal pigment in black) and surrounded by a keratinocyte (plasma membrane in blue, presenting keratin bundles on the cytosol, white arrowheads in left panel). Caveolae

**Fig. 1 Caveolae localization and modulation in human epidermis and 2D co-culture. a** IFM images of melanocytes and keratinocytes co-cultured for 1 day and immunolabelled for Cav1 (caveolin-1) or Cavin1 (top or bottom, respectively; green) and HMB45 (premelanosome protein PMEL, red). Cav1 or Cavin1 polarization, arrowheads; melanocytes, asterisks. The boxed regions mark the zoomed area. Bars, 10 µm. **b** Quantification of Cav1 or Cavin1 mean fluorescent intensity in boxes depicted in **a. c**, **d** Quantification of the frequency of melanocytes with Cav1 or Cavin1 polarization [as in **a**, arrowheads] in mono- or co-cultures (**c**), or maintained in different media (**d**). **e** Conventional 2D EM from human skin tissue fixed chemically (top) or immobilized by high-pressure freezing (HPF, bottom); plasma membranes of keratinocytes (red) and of melanocytes (blue), plasma membrane invaginations with morphological features of caveolae (arrowheads). The boxed regions mark the zoomed area. Bars: (main) 1 µm; (zoom) 100 nm. **f** 3D-model reconstruction electron tomography of the melanocyte–keratinocyte interface in human skin epidermis; plasma membrane of melanocytes (green) and of keratinocytes (blue), limiting membrane of melanosomes (red), melanin (black), caveolae (white) in single (arrowhead) and clustered structures (arrow) and keratin bundles (white arrowheads). See also Supplementary Video 1 and Supplementary Fig. 1f. **g**, **h** Quantification of the number of caveolae [as identified in **e**] per 10 µm of plasma membrane at the indicated interface (**g**) and of individual cell types at melanocyte–keratinocyte interface (**h**) during 3D-HRPE formation. **i** Immunoblot analysis and quantification of Cav1 protein levels (normalized to Calnexin) in melanocytes daily exposed to UV-B radiations (ultraviolet-B; 11 mJ/cm$^2$, 3 days). Asterisk, Cav1 full-length protein and truncated form (upper and lower bands, respectively). Values are mean ± s.e.m. **b** 12 cells. **c**, **d** 150 cells, three independent experiments. **g**, **h** 1 experiment. **i** Three independent experiments. **b** Two-tailed unpaired *t*-test with Welch's correction; **c**, **d** one-way ANOVA with Tukey's multiple-comparison test; **g**, **h** comparison between interface/cells at the same time point: two-tailed unpaired *t*-test with Welch's correction; comparison between time points from the same cell type: one-way ANOVA with Tukey's multiple-comparison test; **i** one-way ANOVA with Sidak's multiple-comparison test. See also Supplementary Table 1.

(white) were observed in the melanocyte as single (Fig. 1f, black arrowhead) or clustered structures known as rosettes (Fig. 1f, arrow) connected to the cell surface[16,35].

3D human reconstructed pigmented epidermis (3D-HRPE) composed of normal human epidermal melanocytes (Mel) and keratinocytes (Ker) are used to study epidermis stratification and pigmentation[36]. The development of the reconstructed tissue includes the initial epidermis stratification at day 4, pigmentation at day 6 and formation of a fully stratified and pigmented epidermis at day 12. To address the distribution and modulation of caveolae during human epidermis development, we specifically examined the two cell–cell interfaces existing in 3D-HRPE; i.e. the melanocyte–keratinocyte and the keratinocyte–keratinocyte interfaces. Representative samples of each day were chemically fixed, thin-sectioned and analyzed by conventional TEM (Fig. 1g, h and Supplementary Fig. 1g, h). From days 4 to 12, the melanocyte–keratinocyte interface showed an increased number of caveolae per 10 µm-length of plasma membrane when compared to homologous keratinocyte–keratinocyte interface (Fig. 1g and Supplementary Fig. 1g). Although the number of caveolae was constant at the melanocyte–keratinocyte interface (Fig. 1g), differences in caveolae density appeared with time for each cell type (Fig. 1h). At day 4, when the tissue stratified, caveolae were fourfold higher in keratinocytes when compared to melanocytes (Fig. 1h). However, from days 4 to 6, when the tissue started to color (Supplementary Table 1), caveolae density increased fivefold in melanocytes (Fig. 1h). As a control, we observed that the number of CCPs, identified by the presence of a characteristic electron-dense coat[37], was similar at both interfaces and cell types and constant over time (Supplementary Fig. 1h, top and bottom panels, respectively). This shows that the density of caveolae in the tissue is higher at the melanocyte–keratinocyte interface than at the keratinocyte–keratinocyte one, and more importantly that such density can vary. Because caveolae numbers increase specifically in melanocytes as compared to keratinocytes when the epidermis acquires its pigmentation, it suggests that caveolae could participate in tissue pigmentation.

Ultraviolet (UV) radiation increases skin pigmentation by stimulating melanocytes to synthesize and transfer the pigment melanin, while modulating the secretion of keratinocyte signaling factors as well as exosomes[7,8]. We thus examined whether daily low doses of UV-B, which mimic physiological solar exposure[8] could modulate the expression levels of Cav1 in melanocytes and keratinocytes (Fig. 1i and Supplementary Fig. 1i, respectively). Cav1 protein levels were increased sevenfold in melanocytes after three consecutive irradiations (Fig. 1i), whereas no difference was observed in keratinocytes (Supplementary Fig. 1i). Thus, UV-B exerts a positive role in modulating Cav1 expression in the epidermal unit, yet more prominently in melanocytes. Altogether, melanocytes modulate the levels and distribution of caveolae in response to extracellular and physiological stimuli, such as keratinocyte-secreted factors and UVs.

**Caveolin-1 regulates cAMP production in melanocytes**. Considering the prominent function of caveolae in intracellular signaling[20] and the significant impact of both keratinocyte-secreted factors and UV-B on caveolae distribution, and Cav1 levels, respectively, we investigated whether caveolae-mediated signaling could contribute to pigmentation in melanocytes. Melanocytes express different receptors that activate signal transduction pathways and increase pigmentation[7,9]. A key signaling molecule in this process is the second messenger cAMP produced by tmACs downstream of GPCR activation[11]. Interestingly, Cav1 and Cav3 can control cAMP production and therefore would contribute to the compartmentalization of this second messenger[23,38]. Melanocytes treated with control siRNA or siRNAs targeting Cav1 (Supplementary Fig. 2a) and grown without any cAMP-stimulating molecules were stimulated with forskolin (FSK; Fig. 2a and Supplementary Fig. 2b), a cell-permeable direct activator of tmACs[39]. Upon FSK stimulation, Cav1-depleted melanocytes increased the intracellular cAMP markedly by 6.5-fold, whereas the increase in cAMP was only threefold in control cells (Fig. 2a). Several studies have reported that caveolae could regulate the activity of various signaling molecules, mostly in an inhibitory fashion and in a process dependent of the caveolin-1 scaffolding domain[40,41] (CSD). Indeed, the catalytic activity of specific tmACs isoforms can be inhibited by a cell-permeable synthetic peptide, which mimics the Cav1 CSD[22], and herein after referred to as CavTratin. The stimulation with FSK of CavTratin-treated melanocytes resulted in a 30% reduction of cAMP intracellular levels (Fig. 2b and Supplementary Fig. 2c). These results strongly suggest that caveolin-1 through its CSD leads to the decrease of the production of cAMP in melanocytes, possibly by reducing the activity of tmACs.

**Caveolin-1 controls pigmentation in melanocytes**. Elevated cAMP levels enhance the activity of protein kinase A (PKA), which phosphorylates the cAMP responsive element binding protein (CREB) and therefore upregulates the transcription of genes associated with pigmentation[11]. Stimulation of cells, either by supplemented medium (containing factors known to promote

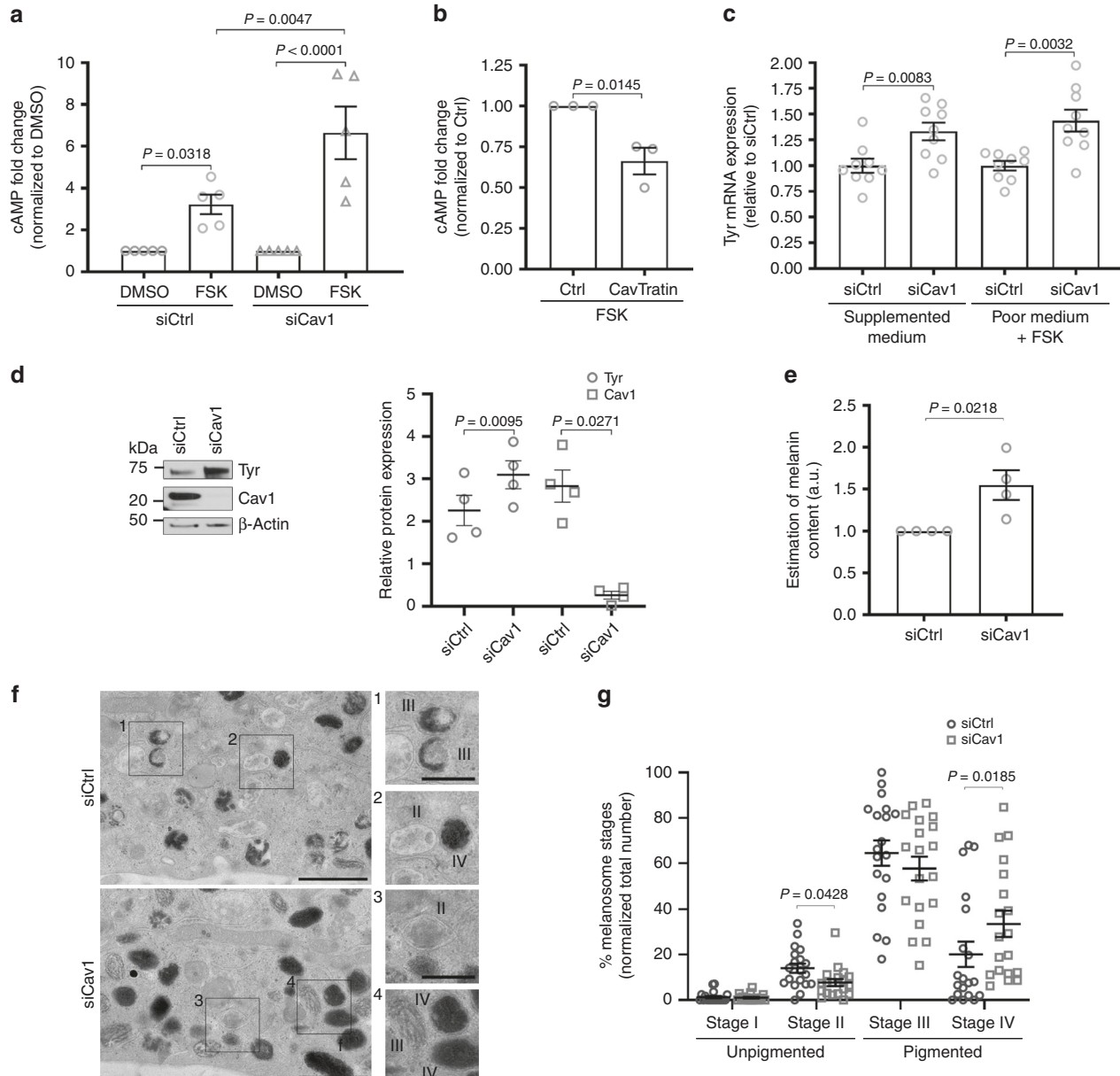

**Fig. 2 Caveolin-1 depletion in stimulated melanocytes increases cAMP production and pigmentation. a**, **b** Quantification of intracellular cAMP fold-change in melanocytes. **a** Melanocytes were transfected with Ctrl (control) or Cav1 (caveolin-1) siRNA for 24 h and incubated with DMSO (dimethylsulfoxide) or 30 µM of FSK (forskolin) for 3 h. **b** Melanocytes were treated with Ctrl (scrambled) or CavTratin (Cav1- scaffolding domain) peptides for 7 h and incubated with DMSO or 30 µM of FSK for 1 h. **c** Quantification of TYR (Tyrosinase) mRNA expression in siCtrl- or siCav1-treated cells maintained in supplemented medium or in poor medium + FSK (3 h) and normalized to GAPDH (glyceraldehyde 3-phosphate dehydrogenase). **d**–**g** Melanocytes treated 5 days with siCtrl or siCav1. **d** Immunoblot analysis of melanocytes treated for 5 days with siCtrl or siCav1 using the indicated antibodies (left) and associated quantifications (right). **e** Estimation of intracellular melanin content normalized to control. **f** Conventional EM images representative of each condition with the respective zooms of the boxed regions; Bar: original 1 µm, zoomed 0.5 µm; II to IV represent different stages of maturation of melanosomes. **g** Quantification of the number of non-pigmented and pigmented melanosomes from EM images as in **f** (*n* = 14 independent cells). Values are mean ± s.e.m. **a** Five independent experiments; **b**, **c** three independent experiments; **d**, **e** four independent experiments. **g** 19 cells, 4 independent experiments. **a**, **g** One-way ANOVA with Holm–Sidak's multiple-comparison test; **b**, **c**, **e** two-tailed unpaired *t*-test with Welch's correction; **d** two-tailed paired *t*-test. See also Supplementary Table 2.

intracellular cAMP production[42,43], Methods) or by FSK, increased the fraction of phosphorylated CREB (p-CREB) similarly in both control and Cav1-depleted melanocytes (Supplementary Fig. 2d). Activation of CREB by phosphorylation leads ultimately to the neo-synthesis of melanin-synthesizing enzymes and increased melanin production[11]. Stimulation of Cav1-depleted melanocytes increased the mRNA expression levels of the rate-limiting enzyme tyrosinase (Tyr, Fig. 2c) and the

L-dopachrome tautomerase[44] (DCT, Supplementary Fig.2e). This was followed by an increase in the levels of both proteins (Fig. 2d and Supplementary Fig. 2f) and in the intracellular melanin content (Fig. 2e). Within melanosomes, synthesized melanin deposits onto a fibrillar matrix that forms on the proteolytic cleavage of the structural protein PMEL[45], and whereby expression levels remained unchanged in Cav1-depleted melanocytes (Supplementary Fig. 2g). Similarly, the mRNA and protein

expression levels of the Rab27a GTPase, which regulates melanosome transport to the cell periphery[5], were constant (Supplementary Fig. 2e, h). These data indicate that Cav1 depletion specifically affects pigment production in melanosomes, but not their structure or their intracellular peripheral localization, as also evidenced by conventional TEM of siCav1-treated melanocytes (Supplementary Fig. 2i). Because pigment production is accompanied by melanosome maturation[4], we used TEM to quantify the early unpigmented melanosomes (stages I and -II) and the mature pigmented melanosomes (stages III and IV) in Cav1-depleted melanocytes. Consistent with the biochemical analyses (Fig. 2d, e), the number of pigmented stage IV melanosomes increased significantly with a concomitant decrease in unpigmented stage II melanosomes in Cav1-depleted melanocytes (Fig. 2f, g and Supplementary Table 2). Taken together, caveolin-1 control early signaling events in melanocytes that affect the transcriptional regulation of melanin-synthesizing enzymes, melanin production and melanosome maturation.

**Caveolae regulate the mechanical response of melanocytes to increased cAMP, to cell–cell contacts and to mechanical stress.** Local production of cAMP at the plasma membrane regulates neuronal cell shape[46], epithelial cell polarity[47] or cell migration[48]. In melanocytes and melanoma cells, the increase of cAMP levels supports dendrite outgrowth[12]. For the last few years, caveolae mechanosensing and mechanoprotective functions have emerged as a new major feature of caveolae in many cell types in vitro and in vivo[25,28]. In this context, caveolae were recently shown to couple mechanosensing with mechanosignaling in human myotubes[29]. Because Cav1 regulates cAMP levels in melanocytes (Fig. 2a, b), we explored the role of caveolae in the mechanical behavior of melanocytes in response to chemical stimulation. Cav1-depleted melanocytes (Supplementary Fig. 3a) were grown in four different media: (1) devoid of stimulating molecules (poor medium), and three others containing stimulating molecules: (2) supplemented medium (Methods), (3) forskolin (poor medium + FSK) or (4) keratinocyte-conditioned media (Ker-CM). The shape of the cells was analyzed using fluorescently labeled phalloidin that stained actin filaments (F-Actin; Fig. 3a). In the absence of signaling molecules (poor medium), control and Cav1-depleted melanocytes preferentially displayed a similar morphology characterized by the presence of at most two protrusions (*; Fig. 3a, b). Interestingly, stimulation of control melanocytes increased the number of protrusions, whereas the majority of Cav1-depleted cells did not yield more than two protrusions (Fig. 3a, b). We then characterized cell morphology by measuring the cell area, major and minor axis and by calculating the length-to-width ratio (Fig. 3c and Supplementary Fig. 3b–d). Without stimulation, the length-to-width ratio was similar in control- and Cav1-depleted melanocytes. After stimulation, only the area of the cell and the minor axis increased in control cells (Supplementary Fig. 3b–d). This caused a slight decrease in the length-to-width ratio (Fig. 3c), which is reflected by cell spreading and formation of dendrite-like protrusions (Fig. 3a, b). Conversely, Cav1-depleted cells responded to stimulation by preserving the cell area (Supplementary Fig. 3b) and by increasing the major axis while decreasing the minor axis (Supplementary Fig. 3c, d), which confirms their elongated shape (Fig. 3a). This marked increase of the length-to-width ratio (Fig. 3c) suggests that cell spreading occurs mainly along the major axis. In addition, time-lapse microscopy of control- or Cav1-depleted-melanocytes cocultured with keratinocytes was performed to investigate the impact of caveolae on the dynamic change of morphology of melanocytes. In the absence of direct cell contact with keratinocytes, control melanocytes responded dynamically by extending

and retracting dendrite-like protrusions along time (Supplementary Video 2). Conversely, Cav1-depleted melanocytes displayed an elongated shape and formed fewer projections (Supplementary Video 3). The difference of response due to the absence of caveolae was better evidenced by delineating the cell boundaries during the 4 h acquisition (Fig. 3d) and consistent with the immunofluorescence microscopy data obtained for stimulated melanocytes in monoculture (Fig. 3a). Therefore, in melanocytes devoid of caveolae, the sole elevation of intracellular cAMP (poor medium+ FSK) or of other cAMP-independent signaling pathways (supplemented medium and Ker-CM) is not sufficient to support the outgrowth of protrusions. Overall, caveolae are required to couple the physical response of melanocytes to extracellular stimuli, more particularly the ones elicited by keratinocytes.

In addition to the established role of extracellular signaling molecules, direct contact between melanocytes and keratinocytes might also promote dendrite outgrowth[49]. As a result, we tested if caveolae contribute to changes in the morphology of the melanocytes in response to cell–cell interactions with keratinocytes. Control melanocytes responded by extending and retracting dendrite-like protrusions when keratinocytes were in close contact (Supplementary Video 4), while Cav1-depleted melanocytes were mostly unresponsive to any contact made by keratinocytes; they formed fewer projections and displayed an elongated shape (Supplementary Video 5). Interestingly, Cav1-depleted melanocytes were more frequently deprived of physical contact by keratinocytes during the total time of acquisition (Fig. 3e). In addition, the frequency of melanocytes–keratinocytes contacts that were long-lasting (1–4 h) decreased (Fig. 3e, Supplementary Videos 4, 5 and Supplementary Table 3). As caveolin-1 has been associated with cell migration[50], we quantified the average velocity and the total traveled distance in control- and Cav1-depleted melanocytes when co-cultured with keratinocytes (Supplementary Fig. 3e). No difference was observed among conditions. Thus, melanocytes devoid of caveolae are unable to produce the outgrowth of protrusions in response to keratinocyte-secreted factors or to direct contact with keratinocytes. Altogether, these data show that caveolae in melanocytes have a key role in melanocyte dendrite outgrowth and the establishment and maintenance of contact with keratinocytes.

Local changes in cell cortex mechanical properties can drive cell contractility and changes in cell shape that are primarily controlled by non-muscle myosin-II motors (NMMIIs)[51]. As the activity of NMMIIs is regulated by the phosphorylation of the regulatory myosin light chain[52] (p-MLC), we investigated the changes in p-MLC levels in melanocytes stimulated with supplemented medium or with FSK (Fig. 3f, g). As compared to control cells, the p-MLC/MLC ratio was decreased in Cav1-depleted melanocytes after stimulation with either supplemented medium or FSK, which suggests that melanocytes devoid of caveolae have a decreased NMMIIs activity. This result indicates that caveolae control the cAMP-dependent changes in shape and dendricity of melanocytes by modulating the contractile force generated by the actomyosin subcortical network.

The mechanical response of cells to changes in shape is also correlated with adjustments in the plasma membrane tension to the cytoskeletal architecture and dynamics[53]. Under mechanical stress, caveolae serve as a membrane reservoir that disassembles rapidly to buffer variations of plasma membrane tension[25]. To address whether the mechanical function of caveolae during the changes in morphology is also reflected in membrane tension variations, we monitored the resistance of the plasma membrane of melanocytes during membrane tension increase induced by hypo-osmotic shock. Melanocytes were pre-incubated with the

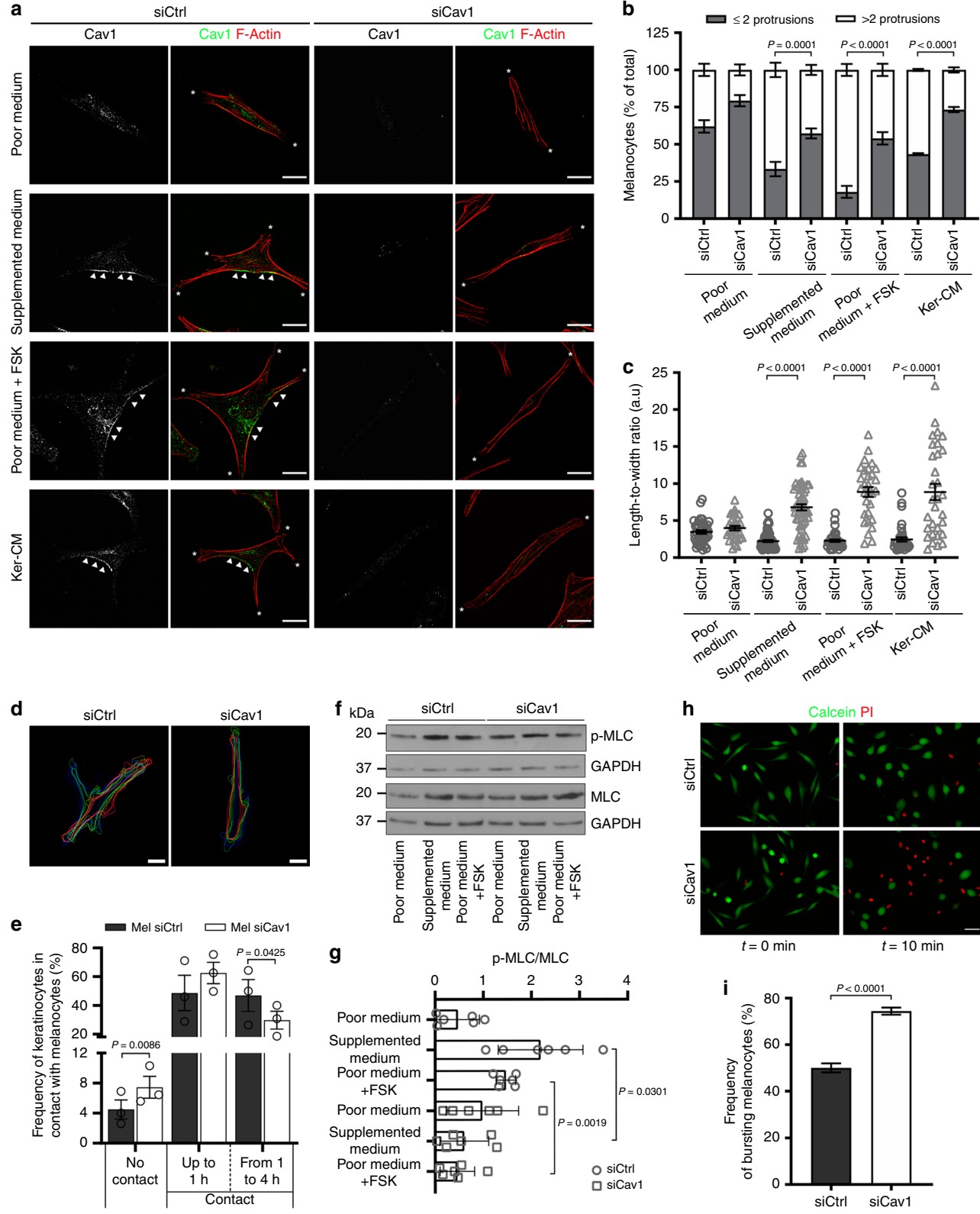

membrane permeant cytoplasmic green-fluorescent dye calcein-AM and exposed to a 30 mOsm hypo-osmotic shock in the presence of propidium iodide (PI), a non-permeant red-fluorescent DNA intercalating agent. A loss of plasma membrane integrity is revealed by a decrease or absence of the calcein-AM signal while acquiring a positive signal for propidium iodide. After 10 min of hypo-osmotic shock, Cav1-depleted melanocytes

(Supplementary Fig. 3f) had burst more frequently than control cells (Supplementary Videos 6, 7 and Fig. 3h, i), confirming that caveolae offer mechanoprotection to melanocytes experiencing membrane tension variations. All in all, these data indicate that caveolae in melanocytes translate extracellular stimuli originating from keratinocytes, such as secreted factors and physical contacts, into a mechanical response of the pigment cell. This is likely to

**Fig. 3 Caveolae contributes to changes in melanocyte morphology, contacts with keratinocytes and mechanoprotection. a** IFM images of melanocytes treated with Ctrl (control) or Cav1 (caveolin-1) siRNA and incubated for 14 h with poor medium (+DMSO), supplemented medium (+DMSO), poor medium + FSK (forskolin, 30 μM) or Ker-CM (keratinocytes-conditioned medium) for 14 h, immunolabelled for Cav1 (caveolin-1, green) and stained for F-actin (phalloidin, red). Cav1 polarization, arrowheads; cell protrusions, asterisks. Bars, 20 μm. **b** Frequency of melanocytes [as in **a**] showing at most two (≤2) or more than two (>2) membrane protrusions (n = 150 cells, three independent experiments). **c** Quantification of the length-to-width ratio of melanocytes cultured as in **a**. **d**, **e** Melanocytes treated for 72 h with siCtrl or siCav1 were co-cultured with keratinocytes for 14 h prior to cell imaging. **d** Representative projection of time-lapse images with interpolated region of interest for the cell's boundaries every 20 min. Bars, 10 μm. See also Supplementary Videos 2 and 3. **e** Frequency of keratinocytes contacting melanocytes for a total of 4 h. **f** Immunoblot analysis of p-MLC (phosphorylated myosin light chain) and MLC (myosin light chain) levels in siCtrl- and siCav1-treated melanocytes maintained for 4 h in poor medium, supplemented medium or poor medium + FSK. **g** Quantification of MLC activation in **f**, corresponding to the ratio of p-MLC on MLC total expression level after GAPDH (glyceraldehyde 3-phosphate dehydrogenase) normalization. **h**, **i** Melanocytes treated with siCtrl or siCav1 for 72 h were incubated with calcein-AM (green) for 15 min, washed and subjected to hypo-osmotic shock (30 mOsm) in the presence of PI (propidium iodide, red) for 10 min. PI-positive cells (red nuclei) indicate melanocytes with ruptured plasma membrane. See also Supplementary Videos 6 and 7. **h** First (0 min) and last (10 min) still images from the time-lapse acquisition. Bars, 50 μm. **i** Frequency of bursting melanocytes. Values are the mean ± s.e.m. **b**, **c**, **e** Three independent experiments. **i** siCtrl, n = 714 cells, siCav1, n = 958 cells; three independent experiments. **g** Six independent experiments. **b**, **c** One-way ANOVA with Tukey's multiple-comparison test; **e** two-tailed paired t-test; **g** one-way ANOVA with Dunnett's multiple-comparison test; **i** two-tailed unpaired t-test with Welch's correction. See also Supplementary Table 3.

occur through the NMMII-dependent control of the actomyosin network and/or by adjusting membrane tension.

**Loss of caveolae impairs melanin transfer in 2D co-culture and 3D-epidermis.** Skin pigmentation relies on the synthesis of the melanin within melanocytes, and its transfer and maintenance within neighboring keratinocytes. Different mechanisms have been proposed for melanin transfer to occur[5,54] and each one requires the local remodeling of the plasma membrane of melanocytes at the near vicinity of keratinocytes. To address the role of caveolae in melanin transfer, siCtrl- and siCav1-treated melanocytes were co-cultured with keratinocytes for 3 days, after which the cells were analyzed by immunofluorescence using HMB45 anti-PMEL antibody[30] (Fig. 4a). Keratinocytes co-cultured with Cav1-depleted melanocytes were less frequently positive for HMB45 staining (Fig. 4b) and, when positive, showed decreased mean fluorescence intensity (Fig. 4c), reflecting in overall that fewer melanin-positive fibrils have been transferred. This result shows that caveolae are required for the efficient transfer of melanin from melanocytes to keratinocytes in co-culture.

Interestingly, the microRNA-203a (miR-203a) downregulates Cav1 expression in melanoma cells[55]. Likewise, normal human melanocytes transfected with the pre-mir-203a showed decreased Cav1 protein expression levels (Supplementary Fig. 4a). When co-cultured with melanocytes treated with pre-miR-203a, melanin transfer had occurred in fewer keratinocytes (Fig. 4a, b) whose the mean fluorescence intensity (HMB45) was decreased (Fig. 4c). The miR-203a is secreted by keratinocytes together with exosomes[8], which suggests that keratinocytes could regulate Cav1 expression levels and caveolae biogenesis in melanocytes in order to control their signaling and mechanical responses and, ultimately, the transfer of melanin.

Finally, we sought to establish the importance of caveolae in pigment transfer in vivo. For this purpose, we used the model of skin epidermis (3D-HRPE) and generated three different types of epidermises composed of normal human keratinocytes either alone (Ker-HRPE) or associated with control- or Cav1-depleted melanocytes. The expression of Cav1 mRNAs was efficiently downregulated after siCav1 treatment in melanocytes (Supplementary Fig. 4b). Macroscopic examination of the reconstructed tissue showed unpigmented epidermis when composed of only keratinocytes, and homogenous pigmented epidermis when control melanocytes were added (Supplementary Fig. 4c; left and middle panels, respectively). In contrast, a non-homogenous pigmentation was observed in the epidermis reconstructed with

siCav1-treated melanocytes (Supplementary Fig. 4c, arrow; right panel). The pigmentation defect was further characterized at the ultrastructural level (Fig. 4d) and revealed that keratinocytes juxtaposed with Cav1-depleted melanocytes contained less melanin than when adjacent to control cells (Fig. 4e and Supplementary Table 4). These data show that caveolae have a novel and critical role in melanin transfer from melanocytes to keratinocytes in the human epidermis.

## Discussion

Pigmentation of the human epidermis represents a natural body photo-protective screen that relies on melanocytes and keratinocytes. To adapt to their environment, like during intense solar exposure, these epidermal cells communicate to orchestrate cellular responses important for producing and disseminating the pigment through the tissue. In this study, we provide evidence for a physiological role of caveolae in human epidermis pigmentation. By exploiting the signaling and mechanical functions of caveolae, melanocytes respond to the extracellular signals sent by keratinocytes to potentiate skin photoprotection. The capacity of caveolae to modulate intracellular signals, to provide mechanoprotection and to support the morphological changes in melanocytes defines them as a molecular platform required for human skin pigmentation.

The asymmetric distribution (or enrichment) of caveolae in melanocytes are positively regulated during the formation of skin, by keratinocyte-secreted factors and by solar mimicking UV-B radiation, indicating that specific plasma membrane domains of melanocytes are preferentially dedicated to the biogenesis, distribution and/or stabilization of caveolae. In addition, the miR-203a secreted together with keratinocytes extracellular vesicles[8] can target Cav1 in both melanoma cells[55] and normal melanocytes. This indicates that extracellular factors secreted by keratinocytes contribute directly to fine-tune Cav1 expression and caveolae in melanocytes, so that cellular responses are organized and coordinated. A downregulation of Cav1/caveolae would promote pigment production in melanocytes whereas an upregulation would favor changes in cell morphology and cell–cell contacts, both leading to melanin transfer and skin pigmentation.

Melanocytes devoid of caveolae have higher production of intracellular cAMP after stimulation, whereas treatment with the Cav1-scaffolding domain (CSD) mimicking peptide, CavTratin, has an opposite effect. A classical view of caveolae function in signaling is associated with the intracellular compartmentalization and concentration of different signaling transduction pathway components[20]. In this context, caveolin-1 could regulate the

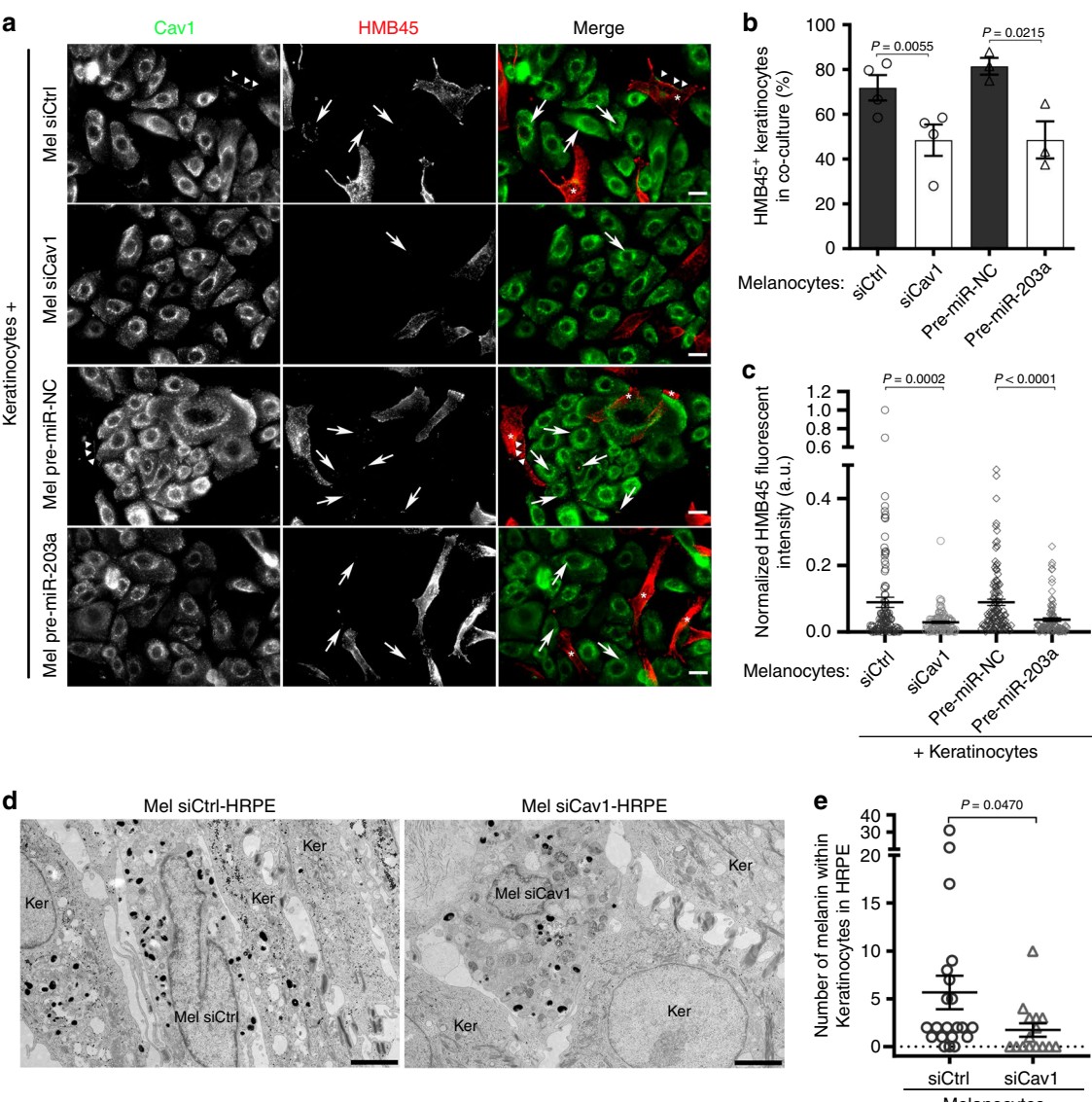

**Fig. 4 Caveolae in melanocytes are necessary for melanin transfer in vitro and in tissue. a–c** Melanocytes treated with Ctrl (control) or Cav1 (caveolin-1) siRNAs, pre-miR-NC (pre-miRNA-negative control) or pre-miR-203a (pre-miRNA-203a) for 5 days were co-cultured with keratinocytes for the last 2 days. **a** IFM images of the co-culture immunolabeled for Cav1 (caveolin-1, green) and HMB45 (premelanosome protein PMEL, red). Keratinocytes positive for HMB45-transferred melanin, arrows; melanocytes in merge panels, asterisks. Bars, 20 μm. **b** Quantification of the frequency of keratinocytes positive for HMB45-positive structures in each condition. **c** Quantification of HMB45 fluorescent intensity in individual keratinocytes positive in **b**. **d** Conventional EM micrographs of 9 days 3D-HRPE (human reconstructed pigmented epidermis) composed of keratinocytes and siCtrl- or siCav1-treated melanocytes. Bars, 2 μm. **e** Quantification of the number of pigmented melanosomes in keratinocytes at the vicinity of melanocytes. Values are the mean ± s.e.m. **b**, **c** Four independent experiments for siCtrl and siCav1 and three independent experiments for pre-miR-NC and pre-miR-203a; **e** one experiment. **b** Two-tailed paired *t*-test; **c**, **e**, two-tailed unpaired *t*-test with Welch's correction. See also Supplementary Table 4.

activity of selected isoforms of tmACs in cells[56,57], whereas the CavTratin peptide could negatively regulate these enzymes in vitro, as reflected by the concomitant decrease of cAMP production after stimulation[22]. So, and even if the mode of action of the CSD is still debated[58], this shows that caveolae mitigate the cAMP-dependent signaling in melanocytes through the possible interaction of Cav1 with tmACs to inhibit their catalytic activity.

In response to increased cAMP production, Cav1-depleted melanocytes display lower NMMIIs activity, which results in diminished contractile forces[51], similarly to Cav1-KO fibroblasts[59]. Consequently, less cortical tension is generated on the plasma membrane, which is necessary for changes in morphology, and may explain the marked failure of melanocytes to extend dendritic-like protrusions in the absence of caveolae. This

strongly suggests that caveolae couple cAMP-induced signaling to the mechanical response of melanocytes. This feature of caveolae might not only be restricted to melanocytes and is likely shared by neural crest-derived cells. Indeed, the modulation of cAMP levels in the vicinity of membrane lipid rafts controls dendritic arborization in mice neurons[60], whereas neuron-targeted Cav1 enhances branching out of the dendrites[61]. Dendrite outgrowth in human melanocytes and murine melanoma cells is also dependent on cAMP[12]. In addition, non-stimulated Cav1-depleted melanocytes seem to display higher p-MLC—yet without impact on the cell shape. That would suggest that caveolin-1 and caveolae might represent key elements modulating the activity of NMMIIs in very different cell types, like melanocytes or fibroblasts[59], and under resting and stimulated conditions.

Endogenous Cav1 and Cavin1, and therefore caveolae, distribute asymmetrically and cell-autonomously in cultured human melanocytes. And polarization of Cav1 and caveolae is observed in different cells during cell migration[50,62]. However and alike immortalized mouse melanocytes[63], normal human melanocytes are poorly motile, which suggests that caveolae asymmetric distribution in these cells should perform functions unrelated to cell migration. In addition to caveolae asymmetry, melanocytes are likely polarized cells because their shape consists of a cell body facing the basal membrane with multiple dendrites extending upwards and because they express proteins specific of epithelial cells[64]. Therefore, we propose that the intrinsic asymmetrical distribution of caveolae imposes a spatial organization of cAMP-dependent pathways and/or downstream targets in melanocytes that contribute to its polarized organization and ensure its cellular functions.

Caveolae are required for two crucial functions in melanocytes: pigment production and transfer. Stimulation of Cav1-depleted melanocytes causes increased cAMP levels and acceleration of pigment production, through the upregulation of Tyrosinase and DCT mRNA and protein expression levels. Pigment synthesis and packaging into melanosomes rely on intracellular signaling pathways, among which cAMP synthesis by tmACs is of key importance[9]. The activation of the GPCR-triggered cAMP pathway increases Tyrosinase and DCT proteins content through increased cell transcriptional activity[11] or post-translational events[43,65]. This indicates that caveolae is key in regulating the production of the pigment through the fine control of cAMP production and downstream pathways.

The fate of melanin in the epidermis is to be transferred to keratinocytes, where it provides color to skin, in addition to shielding the nuclei against UV radiation. Here, we establish a correlation between caveolae formation and human skin pigmentation. Caveolae accumulate at melanocytes when the epidermis becomes pigmented, while impaired caveolae formation in melanocytes, through Cav1 depletion, decreases the transfer of melanin in co-culture and reconstructed epidermis. The dendrites of melanocytes are seen as conduits for melanin transfer and points of contact with keratinocytes and, therefore, their plasticity appears to be important to support these functions. Our results show that caveolae are needed for dendricity of melanocytes, for promoting contacts with keratinocytes and for protecting the plasma membrane of melanocytes against acute rupture after a mechanical stress, thus helping the cells adjust to tension variations. Indeed, both plasma membrane and cortical tension[51] contribute to deform membranes during exo- and endocytosis or changes in cell shape[53,66]. Furthermore, the dynamic cycle of caveolae mechanics, i.e. disassembly and reassembly in response to tension variations that occur during cell morphological changes, could facilitate both dendrite outgrowth and pigment transfer by melanocytes. Nonetheless, the formation of caveolae and non-caveolae Cav1-positive clusters could also exert a spatiotemporal control on melanin secretion by favoring the local remodeling of the plasma membrane in response to signaling cues. Therefore, the coupling of signaling and mechanical responses by caveolae in melanocytes seems important for the regulation of pigment transfer.

Dysregulation of Cav1 expression in the human skin is associated with hyperproliferative diseases such as melanoma and non-melanoma cancers as well as psoriasis[67,68]. In melanoma, Cav1 function remains very controversial, since it is recognized as a tumor suppressor and an oncogene[69]. Such discrepancy might be explained by the variations of Cav1 expression during disease progression, in that the balance between caveolae signaling and mechanical functions in response to the extracellular environment changes during tumor mass growth[70]. Long-term exposure to UV radiation is a key factor causing skin cancers[71] and high levels of expression of the miR-203a occur in psoriatic lesions[72]. We therefore propose caveolae to be an additional modulator of skin pigmentation that couple signaling with mechanical responses in melanocytes. Thus, characterizing the physiology and modulations in response to the extracellular context underlying the functions of caveolae in the skin is a key to decipher their alterations in the disease.

## Methods

**Antibodies**. The following antibodies were used for immunoblot (IB) or immunofluorescence (IFM): rabbit anti-Caveolin-1 (BD Transduction Laboratories; 1:5000 [IB]; 1:200 [IFM]); rabbit anti-PTRF (Cavin1; Abcam; 1:200 [IFM]); mouse anti-HMB45 (recognizing PMEL-positive fibrils onto which melanin deposits, used here as a 'melanin marker'; clone HMB45; abcam; 1:200 [IFM]); mouse anti-α adaptin (clone AP6; abcam; 1:50 [IFM]); mouse anti-Tyrosinase (clone T311; Santa Cruz biotechnology; 1:200 [IB]); mouse anti-DCT (clone C-9; Santa Cruz biotechnology; 1:200 [IB]); rabbit anti-Pep13h[4] (1:200 [IB]); goat anti-Rab27a (SICGEN; 1:1000 [IB]); rabbit anti-phosphorylated CREB (Ser133) (clone 87G3; Cell Signaling technology; 1:1000 [IB]); rabbit anti-CREB (clone 48H2; Cell Signaling Technology; 1:1000 [IB]); mouse anti-phosphorylated MLC2 (Ser19) (Cell Signaling Technology; 1:1000 [IB]); rabbit anti-MLC2 (clone D18E2; Cell Signaling Technology; 1:1000 [IB]); mouse anti-ACTB (β-actin; clone AC-74; Sigma; 1:2000 [IB]); rabbit anti-GAPDH (Sigma; 1:10000 [IB]); rabbit anti-Calnexin (Enzo Life Sciences; 1:1000 [IB]). Secondary antibodies coupled to horseradish peroxidase (HRP) were used at 1:10000 ([IB], Abcam). Secondary antibodies and phalloidin conjugated to 488, 555 and 647 Alexa dyes were used at 1:200 ([IFM], Invitrogen).

**Cell culture**. *Primary cells*: Normal human epidermal melanocytes and normal human epidermal keratinocytes used in this study were isolated from neonatal foreskins and purchased from CellSystems, Sterlab or PromoCell. Melanocytes and keratinocytes were used from passage two and five and maintained in culture in DermaLife Basal Medium supplemented with DermaLife M Life factors (melanocytes-supplemented medium) or in DermaLife Basal Medium supplemented with DermaLife K Life, respectively.

*Cell line*: HeLa cells were cultured in DMEM supplemented with 10% (v/v) FBS, 100 U/ml penicillin G and 100 mg/ml streptomycin sulfate (Gibco). All cells were maintained at 37 °C in a 5% (v/v) $CO_2$ incubator.

**Co-cultures and media incubation**. *Co-cultures*: Melanocytes and keratinocytes or HeLa were seeded in the following ratio, respectively: 1:4 for 24 h before fixation to quantify caveolae asymmetry (Fig. 1); at 1:4 for 14 h before time-lapse acquisition (Fig. 3); and 1:1 for 3 days before fixation to quantify melanin transfer (Fig. 4). All co-cultures were done in melanocyte-supplemented medium.

*Media incubation (IFM and IB)*: Keratinocyte medium from a confluent flask in culture for 48 h was collected and centrifuged at $200 \times g$ to remove cell debris. The Keratinocyte-conditioned medium (Ker-CM) was immediately used or stored at −80 °C (Fig. 1). Melanocytes were seeded and maintained in poor medium (DermaLife Basal Medium without the addition of StiMel8) for at least 3 h after which this medium was removed, the cells washed in phosphate-buffered saline (PBS) and fresh poor medium or poor medium supplemented with 30 μM of forskolin (FSK, Sigma) or with melanocyte-supplemented medium (see above), or Ker-CM was added and kept for ~14 h before fixation for IFM or 15 min to probe for p-CREB/CREB or 4 h to probe for p-MLC/MLC by IB. Dimethylsulfoxide (DMSO, between 0.2 to 0.6%) was added to the medium as a control to FSK addition.

**siRNA and miRNA transfections**. For melanocytes siRNA and miRNA transfections, cells were seeded in the appropriate wells or plates and transfected with 0.2 μM of siRNA using Oligofectamine (Invitrogen) accordingly to manufacturer's instructions using non-targeting siRNA (siCtrl; 5′-AATTCTCCGAACGTGT-CACGT-3′) and siRNA targeting Cav1 (SI00299635 and SI00299628) from Qiagen, or using pre-miR-NC (negative control; #AM17111) and pre-miR-203a (#AM17100) from ThermoFischer Scientific. In 3D-HRPE experiments, melanocytes were transfected previously to reconstruction with 1 μM of siRNA using DharmaFECT and following the manufacturer's protocol (Dharmacon, Horizon) using non-targeting siRNA (Accell non-targeting pool) or siRNA targeting Cav1 (SMARTpool: Accell Cav1) from Dharmacon.

**UV treatment**. Melanocytes and keratinocytes were seeded in six-well plates at day 0 and irradiated with a single shot of 11 mJ cm$^{-2}$ of ultraviolet-B (312 nm) during 3 consecutive days using a Biosun machine (Vilber Lourmat, Suarle´e, Belgium). Cell medium was replaced by PBS before irradiation and replaced by the culture medium just after the treatment. The cells were then incubated overnight and recovered by trypsinization at the indicated time points.

**Skin samples**. Healthy skin samples were obtained from surgical left-over residues of breast or abdominal reduction from healthy women. Written informed consent was obtained in accordance with the Helsinski Declaration and with article L.1243-4 of the French Public Health Code. Given its special nature, surgical residue is subject to specific legislation included in the French Code of Public Health (anonymity, gratuity, sanitary/safety rules etc). This legislation does not require prior authorization by an ethics committee for sampling or use of surgical waste (http://www.ethique.sorbonne-paris-cite.fr/?q=node/1767).

**Human reconstructed epidermis (3D-HRPE)**. The following protocol was adapted from Salducci et al.[73]. Briefly, dead de-epidermized dermis was prepared as follows: Skin samples from healthy adults were obtained, cut in circular pieces (18 mm diameter) and incubated 20 min at 56 °C in HBSS (Invitrogen) containing 0.01% (v/v) Penicillin/Streptomycin (Invitrogen). Epidermis was removed and collected dermis fragments were sterilized in 70° ethanol, washed twice in HBSS, frozen in HBSS (−20 °C) and submitted to six cycles of freezing-thawing to eliminate fibroblasts. The de-epidermized dermis was placed at the bottom of a 6-well plate in 3D-HRPE culture medium composed of IMDM medium (Invitrogen) and keratinocyte medium (CellSystems) at a proportion of 2/3 to 1/3, respectively, and containing 10% (v/v) of calf fetal serum gold (PAA). siRNA-treated melanocytes and non-treated keratinocytes were seeded at a proportion 1:20, respectively, in a culture insert of 8 mm of diameter affixed on the dermis to promote cell adhesion. After 24 h, the culture insert was removed and the de-epidermized dermis submerged for 3 days in 3D-HRPE culture medium to promote cell proliferation. Tissue stratification was initiated by moving up the de-epidermized dermis to the air–liquid interface. All the incubation steps were performed at 37 °C in a 5% CO$_2$ incubator. The number of melanocytes and keratinocytes counted on EM cross-sections of 3D-HRPE provides an estimation of the ratio of the two cells types within the reconstructed tissue. The siCtrl-HRPE consisted of siCtrl-melanocyte:keratinocyte ratio of 1:3.6 (17 siCtrl-melanocytes, 61 keratinocytes); the siCav1-HRPE consisted of siCav1-melanocyte:keratinocyte ratio of 1:2.7 (13 siCav1-melanocytes, 35 keratinocytes).

**Measurement of intracellular cAMP levels**. Melanocytes were transfected once with the indicated siRNAs and maintained in poor medium for 24 h. DMSO or 30 μM of FSK were added to the respective wells for 3 h after which the cells were collected and the intracellular cAMP content measured using the cAMP complete ELISA kit (Enzo Life Sciences) following manufacturer's instructions. Regarding the treatment with the peptides, NHEMs were maintained in poor medium for 14 h before the addition of the peptides Ctrl (scrambled sequence) or CavTratin (Cav1-scaffolding domain, CSD) for 7 h. Then the cells were incubated for 1 h with DMSO or 30 μM of FSK after which the cells were collected and the intracellular cAMP content measured.

**Melanin assay**. Melanocytes were transfected twice at days 1 and 3 for a total of 5 days with the indicated siRNAs. Cells were then collected, sonicated in 50 mM Tris-HCl pH 7.4, 2 mM EDTA, 150 mM NaCl, 1 mM dithiothreitol (with the addition of protease inhibitor cocktail, Roche) and pelleted at 20,000×$g$ for 15 min at 4 °C. The pigment was rinsed once in ethanol:ether (1:1) and dissolved in 2 M NaOH with 20% (v/v) DMSO at 60 °C. Melanin content was measured by optical density at 490 nm (Spectramax 250, Molecular Devices).

**Membrane bursting assay**. Melanocytes were transfected twice with the indicated siRNAs at days 1 and 3 for a total of 3 days and seeded in 12-well plates for 24 h in supplemented medium. At day 4, cells were incubated in 5 μg/ml of Calcein-AM (Life technologies) for 15 min at 37 °C protected from light. The wells were washed once with melanocyte-supplemented medium and maintained until image acquisition. Melanocyte-supplemented medium was diluted in 90% (v/v) water, the equivalent of 30 mOsm hypo-osmotic shock, followed by the addition of 2 mg/ml of propidium iodide (PI, Sigma) and used to induce the rupture of the plasma membrane[29]. Immediately after the medium replacement, images were acquired every minute for a total of 10 min in an inverted microscope (Eclipse Ti-E, Nikon), equipped with a CoolSnap HQ2 camera, using the 20×0.75 NA Plan Fluor dry objective together with MetaMorph software (MDS Analytical Technologies).

**Melanin transfer assay**. The detailed protocol for the melanin transfer assay is described elsewhere[74]. Melanocytes were transfected twice with the indicated siRNA or miRNA at days 1 and 3 for a total of 5 days. At day 3, melanocytes were co-cultured with keratinocytes for a total of 2 days. Images were acquired with an upright epi-fluorescence microscope (Eclipse Ni-E, Nikon) equipped with a CoolSnap HQ2 camera, using a 40×1.4 NA Plan Apo oil immersion objective together with MetaMorph software.

**Immunofluorescence microscopy**. Cell monolayers seeded on glass coverslips were fixed with 4% (v/v) paraformaldehyde in PBS at room temperature for 15 min, then washed three times in PBS and once in PBS containing 50 mM glycine. Primary and secondary antibody dilutions were prepared in the buffer A: PBS containing 0.2% (w/v) BSA and 0.1% (w/v) saponin. The coverslips were

washed once in buffer A and then incubated for 1 h at room temperature (RT) with the primary antibodies. Following one wash step in buffer A, the coverslips were incubated for 30 min at RT with the secondary antibodies. For phalloidin staining, the coverslips were washed in buffer A and incubated overnight in the same buffer with phalloidin at 4 °C. The final wash step was done once in buffer A, once in PBS and once in water. The coverslips were mounted onto glass slides using ProLong™ Gold Antifade Mount with DAPI (ThermoFischer Scientific). Images were acquired on an Applied Precision DeltavisionCORE system (unless stated otherwise), mounted on an Olympus inverted microscope, equipped with a CoolSnap HQ2 camera (Photometrics), using the 40×1.3 NA UPLFLN or the 60×1.42 NA PLAPON-PH oil immersion objectives. Images were deconvolved with Applied Precision's softWorx software (GE Healthcare).

**Time-lapse microscopy**. Melanocytes were transfected twice with the indicated siRNA molecules at days 1 and 3 for a total of 3 days and co-cultured with keratinocytes in an ibidi polymer coverslip μ-slide (Ibidi) for 14 h before imaging. Images were acquired every 5 min for a total of 240 min in an inverted microscope (Eclipse Ti-E, Nikon), equipped with a CoolSnap HQ2 camera, using the 40×0.75 NA Plan Fluor dry objective together with NIS-Elements software (Nikon).

**Electron microscopy**. *Conventional EM*: Human skin epidermis tissues and 3D-HRPE were prepared for EM as described. For high-pressure freezing, the tissue was high-pressure frozen using an HPM 100 (Leica Microsystems) in FBS serving as filler and transferred to an AFS (Leica Microsystems) with precooled (−90 °C) anhydrous acetone containing 2% (v/v) osmium tetroxide and 1% (v/v) of water. Freeze substitution and Epon embedding was performed as described in Hurbain et al.[75]. For chemical fixation, melanocytes seeded on coverslips and transfected twice with the indicated siRNAs at days 1 and 3 for a total of 5 days were fixed in 2.5 % (v/v) glutaraldehyde in 0.1 M cacodylate buffer for 24 h, post-fixed with 1% (w/v) osmium tetroxide supplemented with 1.5% (w/v) potassium ferrocyanide, dehydrated in ethanol and embedded in Epon as described in Hurbain et al.[76]. Ultrathin sections of cell monolayers or tissue were prepared with a Reichert UltracutS ultramicrotome (Leica Microsystems) and contrasted with uranyl acetate and lead citrate.

*Electron tomography*: 300-nm-thick sections were randomly labeled on the two sides with 10 nm Protein-A gold (PAG). Tilt series (2 perpendicular series, angular range from −60° to +60° with 1° increment) were acquired with à Tecnai 20 electron microscope (ThermoFischer Scientific). Projection images (2048 × 2048 pixels) were acquired with a TEMCAM F416 4k CMOS camera (TVIPS). Tilt series alignment and tomogram computing (resolution-weighted back projection) were performed using eTomo[77] (IMOD) software. PAG 10 nm at the surface of the sections was used as fiducial markers. Manual contouring of the structures of interest was performed using IMOD[78].

*Immuno-EM*: Cell samples were fixed with 2% PFA in a 0.1 M phosphate buffer pH 7.4 and processed for ultracryomicrotomy as described in Hurbain et al.[76]. Ultrathin sections were prepared with an ultracryomicrotome UC7 FCS (Leica) and underwent single immunogold labeling with protein-A conjugated to gold particles 10 nm in diameter (Cell Microscopy Center, Department of Cell Biology, Utrecht University). All images were acquired with a Transmission Electron Microscope (Tecnai Spirit G2; ThermoFischer Scientific, Eindhoven, The Netherlands) equipped with a 4k CCD camera (Quemesa, EMSIS, Muenster, Germany).

**Image analysis and quantifications**. *Conventional EM*: Caveolae and clathrin-coated pits[16], and melanosome stages were identified based on their ultrastructural features[4]. Caveolae structures associated with plasma membranes of randomly selected cell profiles were quantified from 2D ultrathin sections of 3D-HRPE. The length of the plasma membranes either of melanocytes or keratinocytes were measured using ITEM software (EMSIS) and the total number of caveolae found associated was reported to 10 μm of plasma membrane of the respective cell type. For melanosome stage quantification, the areas corresponding to the tips of the cells were not considered.

*Immunoblot*: Quantification of protein content on western blot was performed using Fiji software, the background subtracted and intensities were normalized to loading control.

*Caveolae asymmetric distribution by IFM*: Images of endogenous staining for Cav1 and Cavin1 asymmetrically distributed in co-culture were acquired and the background subtracted. Two identical boxes were positioned at the plasma membrane but on opposite sides of the cells and the average fluorescent intensity retrieved. The frequency of Cav1 and Cavin1 asymmetry in melanocytes was defined by identifying cells with one side presenting enriched labeling closely associated with the plasma membrane.

*Protrusions and cell morphology*: A protrusion was defined as an actin-stained extension that originated from the soma of the cell. Isolated cells-treated with siCtrl and siCav1 were selected randomly, imaged and the size parameters (area, length-to-width ratio, major and minor axis) were retrieved. The contour of the cell was achieved by using the wand tool and corrected manually if needed using the tool OR (combine).

*Time of contact*: A cell–cell contact was defined optically when the plasma membrane of keratinocytes and melanocytes were in direct contact, excluding filopodia.

*Cell boundary in time-lapse microscopy*: Representative images: the melanocyte cell contour was drawn manually every 5 frames and in between those frames, the tool Interpolate ROI was used. When needed, the cell boundary was adjusted manually. Regarding the quantification of average velocity and total distance traveled by the cell: segmentation was performed using the "Pixel classification" module of Ilastik software[79], on the original images pre-processed with the "Subtract background" function of ImageJ (rolling ball of 10 pixels). Ilastik classifier was trained to distinguish cell and background (based on the default features: texture/intensity/edge), from manual annotations on a small number of frames, and then applied to the whole movie. Cell masks were computed on ImageJ using the probability maps produced by Ilastik, with a threshold of 0.5, followed by a smoothing as well as basic morphological operations. Manual adjustments could be made when the shape was not correct, using the "Interpolate" tool of the ROI manager. See also Supplementary Code 1.

*Membrane bursting assay*: The background of time-lapse images acquired from the different channels—PI (mcherry) and Calcein-AM (green fluorescent)—was removed with the subtract background tool from Fiji software and the cell's burst determined when the nuclei was red-stained with concomitant loss of fluorescent signal in the green channel at the cytoplasm[29].

*Melanin transfer assay*: Image analysis and quantifications are described elsewhere[74]. All images are maximum-intensity *z*-projections of three-dimensional image stacks acquired every 0.2 μm. Fiji software was used for image analysis.

**Immunoblot**. Cells were collected by trypsinization followed by centrifugation. The cell pellet was resuspended in lysis buffer (20 mM Tris-HCl pH 7.2, 150 mM NaCl, 0.1% (v/v) Triton X-100) containing a protease inhibitor cocktail (Roche). The protein content of the lysates was determined with the Pierce™ BCA Protein Assay Kit (ThermoFischer Scientific), the concentrations adjusted with loading buffer (250 mM Tris-HCl pH 6.8, 10% (v/v) SDS, 50% (v/v) Glycerol, 0.5 M β-mercaptoethanol, 0.5% (w/v) Bromophenol blue) and the samples boiled for 5 min at 95 °C. After SDS-PAGE using NuPage (4–12%) Bis-Tris gels (Invitrogen), the proteins were transferred to 0.2 μm pore-size nitrocellulose membranes (Millipore) and blocked in PBS with 0.1% (v/v) Tween and 4% (w/v) non-fat dried milk or TBS with 0.1% (v/v) Tween and 5% (w/v) BSA. Incubation with primary antibodies was done following the manufacturer's instructions. The detection was done using HRP-conjugated secondary antibodies, ECL Plus Western blotting detection system (GE Healthcare) and exposure to Amersham Hyperfilm ECL (GE Healthcare).

**Quantitative real-time PCR (qPCR)**. *3D-HRPE*: Melanocytes transfected once with the indicated siRNAs for a total of 12 days were collected at days 1 and 12. The RNA was extracted using the Qiagen RNeasy Mini Kit for RNA extraction (Qiagen) and the cDNA generated using the Transcriptor Universal cDNA Master (Roche) following the manufacturer's protocol. 0.3 μg of RNA was used for the quantitative real-time PCR, the mix prepared accordingly to Probes Master (Roche) and the RealTime ready Custom Panels plates (Roche) used for the assay.

*Culture*: Melanocytes were transfected once with the indicated siRNAs for 48 h. Cells were serum-deprived at least 3 h after which RNA was extracted using the RNA extraction kit (MACHEREY-NAGEL) and the cDNA generated from 0.3 μg of RNA using the SuperScript First-Strand Synthesis System (Invitrogen) following manufacturer's protocols. qPCR was performed using the LightCycler 480 SYBR Green I Master (Roche) on plate-based qPCR amplification and detection instrument LightCycler 480 (Roche). GAPDH was used as an endogenous normalizer. See the Supplementary Table 5 for primers sequences. Experiments were performed with at least three biological replicates. The method ΔΔCT was used to obtain the relative expression levels and the ratio between the control and gene of interest was calculated with the formula $2^{-\Delta\Delta CT}$.

**Statistical analysis**. All the statistical analysis on the collected data was performed using GraphPad Prism, version 7 and 8, GraphPad Software, San Diego California, USA (www.graphpad.com). Scored or quantified cells in each experiment were randomly selected, and all experiments were repeated at least three times. Results are reported as mean ± standard error of the mean (s.e.m.). Statistical analysis between three or more experimental groups was performed with one-way ANOVA and Tukey's multiple-comparison test while for comparisons between two sets of data it was used the two-tailed unpaired Student's *t*-test with Welch's correction (unless stated otherwise in figure legends). Differences between datasets were considered significant if $P < 0.05$.

**Reporting summary**. Further information on research design is available in the Nature Research Reporting Summary linked to this article.

## Data availability

All relevant data are available from the authors. The source data underlying the Figs. 1b–d, g–i, 2a–e, a, 3b, c, e–g, i, 4b–c, e and Supplementary Figs. 1h, i, 2a–h, 3a–c, e, f and 4a, b are provided as a Source Data file.

## Code availability

The codes generated during the current study are available on Supplementary Code 1 file.

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

## Acknowledgements

We thank the Structure and Membrane Compartment laboratory, Dr. Mehdi Khaled (Institut Gustave Roussy, Villejuif, France) and Dr. Pablo J. Sáez (Institut Curie, Paris, France) for insightful discussions; Lucie Sengmanivong from the Nikon Imaging Centre at the Institut Curie-CNRS for help in image acquisition; Dr. Gisela D'Angelo (Institut Curie, Paris, France) and Dr. Corinne Bertolotto (Université Côte d'Azur, Nice, France) for critical reading of the manuscript; Dr. Stéphane Elard for collecting and donating tissue samples used for epidermis reconstruction, and Allison Ingalsbe for proof-reading of the manuscript. We acknowledge the Nikon Imaging Center at the Institut Curie/ Centre National de la Recherche Scientifique and the PICT-IBiSA, a member of the France-BioImaging national research infrastructure (ANR-10-INBS-04). This work has received support under the program "Investissement d'Avenir" launched by the French Government and implemented by the Agence Nationale de la Recherche (ANR) with the references ANR-10-LBX-0038 and ANR-10-IDEX-0001-02 PSL, Fondation pour la Recherche Médicale (Equipe FRM DEQ20140329491 Team label to G.R.), Agence Nationale de la Recherche ("MOTICAV" ANR-17-CE13-0020-01 to C.L., and "MYOACTIONS" ANR-17-CE11-0029-03 to C.D.), Fondation ARC pour la Recherche sur le Cancer (PJA20161204965 to C.D., and Programme Labellisé PGA1-RF20170205456 to C.L.). This work received support from the grants ANR-11-LABX-0038, ANR-10-IDEX-0001-02 (Labex CelTisPhyBio to L.D.), Groupe Clarins, Institut Curie, CNRS and INSERM.

## Author contributions

Conceptualization, L.D., C.B., C.G., K.V., G.R., C.L. and C.D.; methodology, L.D., I.H., J.S.-C., N.A., M.D. and C.D.; software, A.-S.M.; formal analysis, L.D., F.G.-M. and J.S.-C.; investigation, L.D., I.H., F.G.-M., J.S.-C., N.A., M.D., M.R., C.V.L. and C.D.; visualization, L.D., I.H. and J.S-C.; writing—original draft, L.D. and C.D.; writing—review and editing, all authors; project administration, C.D.; funding acquisition, G.R., C.L. and C.D.; supervision, L.D., C.B., C.G., K.V., G.R., C.L. and C.D.

## Competing interests

The authors declare no competing interests.
