## [Peer Review File · Nature Communications]

Reviewers' comments:

Reviewer #1 (Remarks to the Author):

In this paper, the authors continue their elegant work on the interplay of melanocytes and keratinocytes in skin pigmentation. They convincingly show, for the first time, that the presence of caveolae in melanocytes significantly enhances transfer of melanin from these cells to keratinocytes. Intriguingly, and consistent with this finding, the authors show that UV-B stimulates expression of caveolin-1 (Cav-1) in melanocytes. (Cav-1 is virtually always directly correlated with caveolae formation in cells; expression of cav-1 is needed for caveolae, and there are no well-established caveolae-independent functions of Cav1.) The authors use both 2D cultures and a more physiological 3D model of skin formation, bolstering the physiological significance of their findings. However, the mechanism by which caveolae act in this process remains more of a puzzle.

The authors look at two phenomena in melanocytes: first, how caveolae modulate cAMP signaling (known to be involved in pigmentation), and second, how caveolae affect dynamic motion and protrusion formation (required for melanin delivery to keratinocytes) in these cells. Surprisingly, the authors find that reducing Cav1 expression with siRNA enhances forskolin-stimulated cAMP production and melanin synthesis (Figure 2). This paradoxical result muddies the waters, suggesting that Cav-1 plays a complex role in regulating pigment formation, that will require further experimentation to clarify.

The experiments in Figure 3, showing that Cav-1 enhances melanocyte motility and protrusion formation, and stimulate contact between co-cultured melanocytes and keratinocytes, are more straightforward. These provide a valuable piece of mechanistic information on how caveolae stimulate skin pigmentation.

The experiments are technically well-performed and the results are believable. The outstanding question in evaluating this manuscript is whether the significance of the findings – without further mechanistic information – is high enough for publication in Nature Communications. The few additional experiments suggested below might clarify some minor outstanding questions, but would not greatly affect the overall significance of the paper.

Specific comments.

Fig. 1a-d. Showing that caveolae are polarized in melanocytes is not in itself an especially novel or significant result. As the authors note, caveolae polarize in a variety of cell types, especially during migration. Sun et al. *J Biol Chem.* 2007 Mar 9;282(10):7232-41 showed that caveolae polarize to the trailing edge of migrating cells; this appears to be true of melanocytes, too, as shown in Figure 3. Thus, this polarization may result simply from the fact that the cells are migrating. The significance of this distribution is not clear.

As detailed below (discussion of Fig. 3), the fact that conditioned medium from keratinocytes increased the percent of melanocytes with polarized caveolae may thus be an indirect piece of evidence that keratinocytes stimulate melanocyte migration. This would be interesting to document directly.

Fig. 1 g, h. The authors measure the total density of caveolae at the melanocyte-keratinocyte interface (that is, the average of the plasma membrane of both cell types in the region of contact), and also the density in the individual cell types at the interface between the cells in 3D co-cultures of melanocytes and keratinocytes. The significance of the first number is not clear, and it's not clear why they made this measurement. There is no reason to think that the density of caveolae in the melanocyte plasma membrane is related to their density in keratinocytes. Caveolae are widely believed to be completely cell-autonomous. The authors examined caveolae density at the cell-cell

interface at various times of differentiation of the 3D culture. Surprisingly, they found that although the density at the interface (average of the two cell types) stayed constant during this process, the density in the individual melanocyte and keratinocyte membrane at the interface zone varied greatly, in a complementary fashion. That is, at Day 4, the density was high in keratinocytes and low in melanocytes, while from Day 4-6, the opposite was true. (The authors omitted the number of cells examined in Fig. 1h; this should be included.) Without further characterization, this remains a puzzling and descriptive observation.

The authors show that the overall caveolae density at the melanocyte-keratinocyte interface was higher than that at the interface between two keratinocytes at all times of differentiation. They conclude that caveolae are "enriched" at the mel-ker interface relative to the ker-ker interface. This is an odd parameter: it would be more useful to compare the density of caveolae in the interface zone of melanocytes with that of "bulk" non-interface melanocyte plasma membrane, to determine if caveolae are enriched in the interface zone. In any event, the significance of any localization of caveolae to the interface zone is completely unclear.

Fig. 3. As noted above, the findings that Cav-1 stimulates melanocyte protrusion formation, dynamism, and interaction with keratinocytes are straightforward than the finding that Cav-1 inhibits cAMP production and melanin synthesis. The authors show that Cav1 is needed for efficient protrusion formation in FSK-stimulated melanocytes grown alone (Fig. 3a-c), and for efficient protrusion formation in melanocytes co-cultured with keratinocytes (Fig. 3d-e). To complement these observations, it would be interesting to directly test the role of keratinocytes in stimulating protrusion formation in melanocytes. That is, the authors could compare spreading/protrusion formation in melanocytes (+/- Cav1 siRNA) when grown alone, versus when co-cultured with keratinocytes (or incubated with conditioned media). This would further support the idea that caveolae were required for physiological response of melanocytes to cues from keratinocytes.

Fig. 3f, showing that caveolae protect melanocytes from hypotonic shock, as has been shown for other cell types, is of low significance. It is not clear that this property is physiologically significant in pigmentation.

In summary, the authors have convincingly shown that caveolae play a role in the interaction of melanocytes and keratinocytes in skin pigmentation. Most impressively, caveolae are clearly shown to enhance pigment transfer from melanocytes to keratinocytes (Fig. 4). Since pigment transfer occurs from the end of melanocyte protrusions, it is reasonable that caveolae act at least in part by stimulating protrusion formation and melanocyte-keratinocyte contact, as shown in Fig. 3. The significance of the ability of caveolae to inhibit cAMP formation is less clear. Although in-depth mechanistic characterization of this process remains for the future, the observations presented here represent an intriguing first step.

Reviewer #2 (Remarks to the Author):

In this manuscript the authors utilized a wide range of molecular and cell biology working models and modern technologies to characterize the role of caveolae in skin pigmentation. They demonstrate caveolae polarize in melanocytes and are particularly abundant at melanocyte-keratinocyte interface; depletion caveolae by using RNAi in melanocytes increased intracellular cAMP level and melanin pigments, but decreases cell protrusions, cell-cell contacts, pigment transfer and epidermis pigmentation. In addition, they also show caveolae in melanocytes can be regulated by ultra-violet radiations and miR-203a secreted from keratinocytes. These data first-time establish caveolae as a key mediator for the physiological regulations between melanocytes and keratinocytes. This is a novel study with comprehensive data sets that strongly support the conclusions. It will help the field to establish the physiological roles caveolae in skin biology.

Major question: The roles of caveolae in melanocytes are broadly related to signal transduction (cAMP), cell-cell contact, and mechanical sensing. It looks like three all play significant roles in melanocytes functions and pigmentation, which are well supported by the data. However, some of them are not convincing. For example, in cAMP signaling, the authors state that "... the production of cAMP in melanocytes through direct binding to the Cav1-CSD." Although Cav1 CSD mimics peptide has effect, it doesn't mean this is through a native cav1 CSD direct binding mechanism. Considering previous report (Dev Cell. 2012, 17;23(1):11), other possibilities may exist, and this need to be further discussed.

Minor: Fig 1i, it states "... n=2 independent experiments". How is the p-value or SE calculated?

Reviewer #3 (Remarks to the Author):

In this manuscript, Domingues et al. investigate the role of caveolae in melanocyte-keratinocyte communication and pigmentation. The authors observed an accumulation of caveolae proteins in melanocytes cocultured with keratinocytes. They depleted core structural components of caveolae in primary melanocytes and reported altered melanocyte morphology and hyperpigmentation compared to control cells. Furthermore, Cav1-depleted melanocytes were more susceptible to bursting upon hypo-osmotic shock, suggesting an altered membrane integrity when caveolae are dysfunctional. The data presented in this study are interesting and provide evidence of caveolae as part of the skin pigmentation process. However, several findings are over interpreted and currently not sufficiently connected, and the manuscript title is not supported by the data presented. The findings on Cav1 function in cAMP signaling and melanocyte morphology do not strengthen the authors' claims, as neither have been linked to skin pigmentation. It remains open, in which step of skin pigmentation (sensing of pigmentation inducing signals, melanin production, melanosome packaging, melanosome transport, dendrite development, melanocyte-keratinocyte contact, melanin transfer) caveolae are important.

Major points:

1. The accumulation of Cav1 and Cavin in melanocytes cocultured with keratinocytes is an interesting finding. How do caveolae proteins distribute in melanocytes equally surrounded by keratinocytes? What do the authors mean with the term "polarized" in this context? It seems surprising that caveolae do not accumulate at the tips of the dendrites, where melanin transfer mainly occurs but rather at perinuclear sides and at the lateral sides.
2. From conditioned-media experiments, the authors conclude that asymmetric caveolae distribution is caused by keratinocyte-secreted factors. Mechanistically, such result would be very interesting. The methods section, however, suggests that with the indicated melanocyte medium an inappropriate control was used. The keratinocyte-conditioned medium contains keratinocyte medium supplements (DermaLife K Life), whereas the melanocyte medium contains Dermalife M life supplements. Thus any differences observed in melanocytes incubated with either melanocyte medium or keratinocyte-conditioned (keratinocyte) medium are not necessarily caused by secreted keratinocyte-derived factors but could also be due to the difference in the commercial media supplements. The authors should use the same medium for both cell types or provide essential negative controls to justify their claim.
3. The authors provide evidence that Cav1 controls cAMP signaling, and propose that this is an upstream event of controlling pigmentation in their system. However, cAMP is involved in many signaling pathways, not only in pigmentation, and no data are provided that altered cAMP levels indeed cause altered melanin synthesis in their system. This part requires further mechanistic exploration to test whether Cav1 controls pigmentation through cAMP regulation.
4. A key finding of this manuscript is the involvement of caveolae proteins in pigmentation. Late stage melanosomes and melanin synthesis proteins like tyrosinase are enriched upon loss of Cav1. The authors claim that caveolae control pigmentation via cAMP, which as stated has not been demonstrated. Instead, various other data in this manuscript together suggest that the primary

defect of Cav1 depletion is an inability to release pigment (e.g. more stage IV melanosomes, less pigment transferred to keratinocytes). Thus it seems possible that Cav1 promotes the transfer rather than the initial synthesis of melanin. To dissect the mechanism of Cav1-mediated pigment regulation the authors could determine if transcription of melanin synthesis genes is altered by forskolin treatment and Cav1 depletion. If Cav1 indeed acts through cAMP one would expect this. If, however, the Tyr and DCT increase is only detectable on protein level, this might argue for an indirect effect caused by accumulation of late stage melanosomes within cells and hence associated protein machineries. This could perhaps also explain why the authors did not detect differences in Rab27a levels.

5. The title and abstract suggest that Caveolae control melanocyte mechanics. This, however, has not been demonstrated, hence the terminology used throughout the manuscript is misleading. In figure 3a-e, morphological changes in Cav1-depleted melanocytes are presented. The authors call this consistently "mechanical behavior" although no mechanical parameters have been assessed. Though the hypo-osmotic shock assay is closer to probing potential differences in membrane tension, it also bears limitations given the lack of knowledge how siRNA-mediated depletion of Cav1 affects the overall viability. In fig. 3f, t=0, siCav1 cells seem more round and possibly more PI-positive, but no quantitative data are provided for this time point. A careful characterization of proliferation and apoptosis following Cav1 depletion is required, as a higher susceptibility to cell death could result in a less protrusion-rich morphology and higher number of PI-positive cells, which the authors now attribute to mechanical defects. Similarly, the altered morphology might be a consequence of altered cell-matrix adhesion. Independent of above limitations, the current morphological data have not been properly connected to the main question: how does Cav1 promote skin pigmentation? One may consider leaving this whole set out.

6. The authors used the 3D-HRPE model and observed reduced melanin transfer in Cav1-depleted melanocytes. However, an overall reduced number of melanocytes could as well explain reduced melanin content in keratinocytes during the establishment of the 3D-HRPE model. The depigmented spot in the middle of siCav1-HRPE seems surprising but was not explained. It might be a hint for lower numbers of melanocytes. The number of melanocytes should be examined e.g. in HRPE cross-sections, and the nature of the pigmentation pattern should at least be discussed. Moreover, quantitative data are required.

Minor points:

1. The manuscript text would benefit from proof-reading. Often plural forms were used where singular would be preferred. For instance, the title should read: "Caveolae coupling of melanocyte signaling and mechanics is required for...", or "keratinocyte-secreted".
2. In many blots, the authors show bar diagrams, representing only the mean and SEM. All single data points should be displayed. Moreover, number of replicates should be stated in each legend, unambiguously for each data set.
3. The gene PMEL encodes a melanocyte-specific type 1 transmembrane glycoprotein and is not equal to melanin. For clarity, the authors should strictly use the name of the antibody/antigen in Fig. 1a, Suppl. Fig.1a and Fig.4a.
4. The data in fig. 2f are displayed as percentage. An absolute increase in stage IV melanosomes would result in relative lower percentage of stage II melanosomes. How do the absolute numbers of melanosomes at different stages depend on Cav1 expression? Is there really a reduction of immature melanosomes?
5. The micrographs in Suppl. Fig 1a do not reflect asymmetric distribution of caveolae proteins in the melanocyte monoculture.
6. Why does UV-B treatment suppress the expression of Cav1 and Cavin at day 1 before it increases at day 2 and day 3? The authors should at least discuss this point.
7. Fig.1e and Suppl. Fig.1d show the same micrograph. This is not considered scientifically sound.
8. The y axis descriptions in Fig.2a & 2b should state "cAMP fold change"
9. In line 246-247 in the main text authors write "..., the major axis increased, while the minor axis increased", which should be corrected in "... minor axis decreased"
10. In line 248 in the main text authors refer to Fig.3d, which should be corrected to Fig.3c

11. The micrographs in Fig.4a do not reflect asymmetric Cav1 distribution in the melanocytes.
12. The y axis legend in Fig.4c does not explain to what the fluorescent intensity is normalized to.

Point-by-point answer to the Reviewers

Reviewers' comments:

Reviewer #1 (Remarks to the Author):

In this paper, the authors continue their elegant work on the interplay of melanocytes and keratinocytes in skin pigmentation. They convincingly show, for the first time, that the presence of caveolae in melanocytes significantly enhances transfer of melanin from these cells to keratinocytes. Intriguingly, and consistent with this finding, the authors show that UV-B stimulates expression of caveolin-1 (Cav-1) in melanocytes. (Cav-1 is virtually always directly correlated with caveolae formation in cells; expression of cav-1 is needed for caveolae, and there are no well-established caveolae-independent functions of Cav1.) The authors use both 2D cultures and a more physiological 3D model of skin formation, bolstering the physiological significance of their findings. However, the mechanism by which caveolae act in this process remains more of a puzzle.

The authors look at two phenomena in melanocytes: first, how caveolae modulate cAMP signaling (known to be involved in pigmentation), and second, how caveolae affect dynamic motion and protrusion formation (required for melanin delivery to keratinocytes) in these cells. Surprisingly, the authors find that reducing Cav1 expression with siRNA enhances forskolin-stimulated cAMP production and melanin synthesis (Figure 2). This paradoxical result muddies the waters, suggesting that Cav-1 plays a complex role in regulating pigment formation, that will require further experimentation to clarify.

The experiments in Figure 3, showing that Cav-1 enhances melanocyte motility and protrusion formation, and stimulate contact between co-cultured melanocytes and keratinocytes, are more straightforward. These provide a valuable piece of mechanistic information on how caveolae stimulate skin pigmentation.

The experiments are technically well-performed and the results are believable. The outstanding question in evaluating this manuscript is whether the significance of the findings – without further mechanistic information – is high enough for publication in Nature Communications. The few additional experiments suggested below might clarify some minor outstanding questions, but would not greatly affect the overall significance of the paper.

We thank the reviewer for the insightful assessment of our manuscript. We are glad that the reviewer has found our work “elegant, technically well-driven and convincing”. We hope that our answers, text modifications and additional data will address the reviewer’s concerns.

Specific comments.

1-Fig. 1a-d. Showing that caveolae are polarized in melanocytes is not in itself an especially novel or significant result. As the authors note, caveolae polarize in a variety of cell types, especially during migration. Sun et al. J Biol Chem. 2007 Mar 9;282(10):7232-41 showed that caveolae polarize to the trailing edge of migrating cells; this appears to be true of melanocytes, too, as shown in Figure 3. Thus, this polarization may result simply from the fact that the cells are migrating. The significance of this distribution is not clear.

As detailed below (discussion of Fig. 3), the fact that conditioned medium from keratinocytes increased the percent of melanocytes with polarized caveolae may thus be an indirect piece of evidence that keratinocytes stimulate melanocyte migration. This would be interesting to document directly.

We agree that it has been reported that a number of cell types display caveolae polarization. However, and to our knowledge, our data represent the first description that melanocytes do

polarize caveolae and in a process that does not rely on the migration of melanocytes (see below). In that sense, we believe that the data is novel.

While melanoblasts (the precursor of melanocytes) and melanoma cells (malignant transformed melanocytes) can display a migratory phenotype, normal differentiated epidermal melanocytes are poorly motile cells in 2D *in vitro* systems as well as in tissue (Bonaventure et al., 2013); yet with few exceptions (e.g. skin wound repair; (Keswell et al., 2015)). For instance, WT immortalized mouse melanocytes were very poorly motile cells in 2D (~0,3 $\mu\text{m}/\text{min}$; (Gallagher et al., 2013)), which do not support an obvious directed motion *in vitro* (as we had already discussed, now **page15/ line 391** of the revised main manuscript).

As mentioned by the reviewer, the polarization of caveolae has been associated to cell migration, as observed in mouse embryonic fibroblasts (MEFs) and some transformed cells. For instance, Caveolin1 (Cav1) KO MEFs displayed an increased average velocity (1 $\mu\text{m}/\text{min}$) as compared to WT MEFs (0,8 $\mu\text{m}/\text{min}$) (Grande-García et al., 2007). We have investigated the migratory phenotype of the normal human epidermal melanocytes, depleted or not for Cav1, by tracking them over time (for a total of 4 hours) in conditions where they are grown together with keratinocytes.

First, we show in the new **Supplementary Fig.3e** that the average instantaneous speed (~0,2 $\mu\text{m}/\text{min}$) of normal human melanocytes as well as the total travel distance are similar in control- and Cav1-depleted conditions (siCtrl: 216.7 \pm 26.3 μm ; siCav1:197.0 \pm 23.7 μm). Of note, the average velocity of control or Cav1-depleted human normal melanocytes is comparable to the average speed of immortalized WT mouse melanocytes (Gallagher et al., 2013), but almost 5 times slower than the one measured in Cav1 KO MEFs (Grande-García et al., 2007). Therefore, **Cav1 expression, and so caveolae, do not contribute to the 2D *in vitro* migration of human normal melanocytes.**

Together, **these observations lead us to exclude that the polarization of caveolae (submitted Fig.1) supports the migration of melanocytes or is the result of their migration** (see note **page 11, I277** and **Supplementary Fig.3e**). Consequently (and in association with the other presented data in the revised manuscript), these results suggest that the asymmetrical distribution of caveolae supports other physiological functions of the melanocyte (e.g. cell signaling, shape and pigmentation).

Bonaventure, J., M.J. Domingues, and L. Larue. 2013a. Cellular and molecular mechanisms controlling the migration of melanocytes and melanoma cells. *Pigment Cell Melanoma Res.* 26:316–325. doi:10.1111/pcmr.12080.

Gallagher, S.J., F. Rambow, M. Kumasaka, D. Champeval, A. Bellacosa, V. Delmas, and L. Larue. 2013. Beta-catenin inhibits melanocyte migration but induces melanoma metastasis. *Oncogene.* 32:2230–2238. doi:10.1038/onc.2012.229.

Grande-García, A., A. Echarrí, J. de Rooij, N.B. Alderson, C.M. Waterman-Storer, J.M. Valdivielso, and M.A. del Pozo. 2007. Caveolin-1 regulates cell polarization and directional migration through Src kinase and Rho GTPases. *J. Cell Biol.* 177:683–694. doi:10.1083/jcb.200701006.

Keswell, D., S.H. Kidson, and L.M. Davids. 2015. Melanocyte migration is influenced by E-cadherin-dependent adhesion of keratinocytes in both two- and three-dimensional *in vitro* wound models. *Cell Biol. Int.* 39:169–176. doi:10.1002/cbin.10350.

2-Fig. 1 g, h. The authors measure the total density of caveolae at the melanocyte-keratinocyte interface (that is, the average of the plasma membrane of both cell types in the region of contact), and also the density in the individual cell types at the interface between the cells in 3D co-cultures of melanocytes and keratinocytes. The significance of the first number is not clear, and it's not clear why they made this measurement. There is no reason to think that the density of caveolae in the melanocyte plasma membrane is related to their density in keratinocytes. Caveolae are widely believed to be completely cell-autonomous.

We made this measurement to bring the first evidences that caveolae could play particular functions in skin biology, by defining whether or not the density of caveolae in the human epidermis can differ according to the cell-cell interface in the skin. We chose the Melanocyte-

Keratinocyte and Keratinocyte-Keratinocyte interfaces because they are the only cell-cell interactions found in the reconstructed 3D pigmented epidermis (that consists only of melanocytes and keratinocytes). We thus consider that the data presented is significant (**Fig.1g**) because it shows that the density of caveolae is higher at the Melanocyte-Keratinocyte interface than at the Keratinocyte-Keratinocyte interface. This reveals that caveolae are non-homogenously distributed throughout the cell-cell interfaces found in the epidermis and suggests that melanocytes and/or keratinocyte would exploit caveolae for skin functions. A note has been added in **page 7, I148**.

Then, we agree with the reviewer that “*Caveolae are widely believed to be completely cell-autonomous*”. We showed (**Fig.1c**, light grey bars) that 1/3 of the melanocyte population (when grown alone) displayed an asymmetric distribution of caveolae, illustrating a cell autonomous feature. Importantly, we measured a very significant increase (x2) in the number of melanocytes harboring an asymmetric distribution of caveolae, when cultured together with keratinocytes (**Fig.1c**, white bars) or when incubated with keratinocytes-conditioned medium (Ker-CM; **Fig.1d**, white bars). This result demonstrates that this particular distribution of caveolae in melanocytes can be enhanced via a non-cell autonomous (keratinocyte-derived) process. In addition, the identification that the miR203a (secreted by keratinocytes (Lo Cicero et al., 2015)) suppressed the expression of Cav1 in melanocytes (our study, **Supplementary Fig.4a**) and in melanoma cells (Conde-Perez et al., 2015) indicated that keratinocytes can alter the amount of caveolae present in melanocytes and/ or the redistribution of Cav1 to the plasma membrane. Finally, and **even if caveolae distribution in melanocyte is largely cell autonomous, our result demonstrates that keratinocytes secrete factors can modulate the number and distribution of caveolae in melanocytes**. We have added a note (**page 14, I358-367**) to clarify this point.

Conde-Perez, A., G. Gros, C. Longvert, M. Pedersen, V. Petit, Z. Aktary, A. Viros, F. Gesbert, V. Delmas, F. Rambow, B.C. Bastian, A.D. Campbell, S. Colombo, I. Puig, A. Bellacosa, O. Sansom, R. Marais, L.C. Van Kempen, and L. Larue. 2015. A caveolin-dependent and PI3K/AKT-independent role of PTEN in beta-catenin transcriptional activity. *Nat. Commun.* 6:8093. doi:10.1038/ncomms9093.

Lo Cicero, A., C. Delevoeye, F. Gilles-Marsens, D. Loew, F. Dingli, C. Guere, N. Andre, K. Vie, G. van Niel, and G. Raposo. 2015. Exosomes released by keratinocytes modulate melanocyte pigmentation. *Nat. Commun.* 6:7506. doi:10.1038/ncomms8506.

3-The authors examined caveolae density at the cell-cell interface at various times of differentiation of the 3D culture. Surprisingly, they found that although the density at the interface (average of the two cell types) stayed constant during this process, the density in the individual melanocyte and keratinocyte membrane at the interface zone varied greatly, in a complementary fashion. That is, at Day 4, the density was high in keratinocytes and low in melanocytes, while from Day 4-6, the opposite was true. (The authors omitted the number of cells examined in Fig. 1h; this should be included.) Without further characterization, this remains a puzzling and descriptive observation.

The number of cell profiles examined in the new **Fig.1h** has been now included in the revised manuscript (see **Supplementary Table 1, page 38**).

We agree with the reviewer that this quantification is descriptive. But, we are not aware of another experimental approach that would provide further characterization of caveolae distribution in physiologically relevant 3D-skin models. Pigmented reconstructed epidermis has to be analyzed at various times of reconstruction and at the ultrastructural level to morphologically identify caveolae. Though descriptive, we consider this observation of interest and of relevance in the context of our study. We believe that the demonstration of the role of caveolae during the pigmentation of melanocytes (**Fig.2** and **Supplementary Fig.2**), the transfer of melanin and overall tissue pigmentation (**Fig.4**) are in very good agreement with

the increased number of caveolae at the melanocyte-keratinocyte interface at the time of tissue pigmentation (Day6).

The **Fig.1h** described the density of caveolae in the individual melanocytes and keratinocytes at the interface zone. It has to be noted that the number of caveolae at the plasma membrane of melanocytes is statistically different from the number of caveolae at the plasma membrane of keratinocytes during the time required for epidermis construction and pigmentation (day 4 and day 6; see **Supplementary Table 1**, panels g and h) (**Fig.1h**). In addition, **the difference in the number of caveolae profiles at the plasma membrane of melanocytes between day 4 and day 6 of tissue reconstruction is also statistically significant**. A brief note has been added on **page 7, I157-162**.

4-The authors show that the overall caveolae density at the melanocyte-keratinocyte interface was higher than that at the interface between two keratinocytes at all times of differentiation. They conclude that caveolae are “enriched” at the mel-ker interface relative to the ker-ker interface. This is an odd parameter: it would be more useful to compare the density of caveolae in the interface zone of melanocytes with that of “bulk” non-interface melanocyte plasma membrane, to determine if caveolae are enriched in the interface zone. In any event, the significance of any localization of caveolae to the interface zone is completely unclear.

The reviewer proposes to compare the density of caveolae at the plasma membrane of melanocyte facing keratinocyte with the one facing no keratinocyte (referred as “bulk”). The reconstructed epidermis is made of normal melanocytes and keratinocytes seeded on top of a dead dermis (see Methods section) and cultured at the air/liquid interface. Therefore, and due to the low number of melanocytes as compared to keratinocytes (alike to the proportion described in the normal human epidermis; approximately 1 melanocyte to 40 keratinocytes), the plasma membrane of a melanocyte is almost exclusively facing the one of a keratinocyte. Therefore, and due to the architecture of the reconstructed epidermis, we have restricted our analysis to the melanocyte-keratinocyte interface. But we have still decided to compare it with the other abundant cell-cell contact area found in the epidermis, i.e. the keratinocyte-keratinocyte interface (**page 7, I148**).

Based on the reviewer concern, we understand that our conclusion could sound overstated, and that the word “enriched” might be confusing. We have removed this word and **we referred now factually to that observation by stating that the density of caveolae at the melanocyte-keratinocyte interface is higher than at the keratinocyte-keratinocyte one** (**page 7, I161**).

5-Fig. 3. As noted above, the findings that Cav-1 stimulates melanocyte protrusion formation, dynamism, and interaction with keratinocytes are straightforward than the finding that Cav-1 inhibits cAMP production and melanin synthesis. The authors show that Cav1 is needed for efficient protrusion formation in FSK-stimulated melanocytes grown alone (Fig. 3a-c), and for efficient protrusion formation in melanocytes co-cultured with keratinocytes (Fig. 3d-e). To complement these observations, it would be interesting to directly test the role of keratinocytes in stimulating protrusion formation in melanocytes. That is, the authors could compare spreading/protrusion formation in melanocytes (+/- Cav1 siRNA) when grown alone, versus when co-cultured with keratinocytes (or incubated with conditioned media). This would further support the idea that caveolae were required for physiological response of melanocytes to cues from keratinocytes.

We agree with the reviewer and we have performed the proposed experiments in condition in which siCtrl and siCav1-treated melanocytes were incubated with keratinocyte-conditioned media (Ker-CM). We have now included additional immunofluorescence data (new **Fig.3a**) and associated quantifications of the number of protrusions (new **Fig.3b**), the description of

the shape (length-to-width ratio, new **Fig.3c**) and the area, the major and minor axes of the cells (new **Supplementary Fig.3b-d**).

Briefly and as compared to siCtrl cells, the new results show that siCav1 melanocytes elongate more and form less protrusions when incubated in keratinocyte-conditioned media. These results are similar to the ones that were originally obtained with siCav1 melanocytes grown in their supplemented medium or in poor media supplemented with FSK. **Therefore, our results show that caveolae are required for the physiological response of melanocytes (e.g. cell spreading, protrusion formation) to cues from keratinocytes.** A note describing the new results has been included (**page 10, I236-254 and p11, I262-266**).

6-Fig. 3f, showing that caveolae protect melanocytes from hypotonic shock, as has been shown for other cell types, is of low significance. It is not clear that this property is physiologically significant in pigmentation.

As stated by the reviewer, caveolae have now been shown to protect various cell types from hypotonic shock. However, it has never been demonstrated in melanocytes. Therefore, the demonstration of the protective role of caveolae in melanocytes can be considered of importance in the context of our study that defines the roles of caveolae during melanocyte and skin pigmentation.

The experimental setup (hypo-osmotic shock) used here is by default not physiological, but the obtained results show very likely that the absence of caveolae in melanocytes leads to a change of their plasma membrane tension, which is reflected by an increased fragility and more susceptibility to rupture. As mentioned by referee#3 (*"the hypo-osmotic shock assay is closer to probing potential differences in membrane tension"*), this data leads us to propose that **caveolae is required to protect the plasma membrane of melanocytes when they experience mechanical stress or membrane remodeling.**

Considering the comment of the reviewer, we have now better examined the impact of caveolae on melanocytes mechanics. We have investigated whether the activity of the non-muscle myosins-II (NMMIIs) was required for melanocytes morphology. The rationale is the following. Changes in cell shape are mainly driven by the plasma membrane-associated subcortical actomyosin network. There, myosins and especially NMMIIs generate contractile forces that create a tension translated into plasma membrane deformations (see a recent review from Ewa Paluch's lab, (Chugh and Paluch, 2018)). The cortical tension can be thus modulated by the activity of myosins-II that is largely associated with the phosphorylation status of the regulatory Light Chain (MLC). Therefore, many cellular studies report the phosphorylation of MLC (p-MLC) as an indicator of NMMIIs activity and thus as a read-out of the contractile force generated by NMMIIs (Heissler and Manstein, 2013).

Cav1-depleted melanocytes stimulated by forskolin or with the supplemented medium showed less MLC phosphorylation (p-MLC) as compared to control (see new **Fig.3f-g**). So, it suggests that **less contractile forces were generated in absence of caveolae, leading to a lower cortical tension of the plasma membrane of the Cav1-depleted melanocytes.** It has to be noted that such influence of Cav1 expression on p-MLC is not restricted to melanocytes because it was already reported in Cav1-KO fibroblasts (Goetz et al., 2011).

In conclusion, Cav1-depleted melanocytes display lower NMMIIs activity in response to increased cAMP production (see new **Fig.3f-g**). This would result in diminished contractile forces and, consequently, less cortical tension generated on the plasma membrane. Together with the hypo-osmotic shock experiment (**Fig.3h-i**), it suggests that melanocytes devoid of caveolae displayed a lower contractility as compared to control cells. Given that a proper adjustment of the cell contractility is most likely necessary for changes in cell morphology, this may explain the dramatic failure of melanocytes to extend dendritic-like protrusions in the absence of caveolae (**Fig. 3a-b**).

Taken together, the new data suggests that **Cav1 and caveolae control the cAMP-dependent changes in shape and dendricity of melanocytes by modulating the contractile force generated by the actomyosin subcortical network. Such force is likely relying on the activity of the NMMIIs and is needed to protrude the plasma membrane of melanocyte in response to signaling.**

We have described the new data (**pages 11-12, I285-294 and page 12, I311**) and further discussed them (**page 15, I378; page 16, I418**). We believe that the additional data included in the revised manuscript strongly reinforce the role of caveolae in melanocytes mechanics and better couple this mechanical aspect to the control of signaling by caveolae in melanocytes during pigmentation.

Chugh, P., and E.K. Paluch. 2018. The actin cortex at a glance. *J. Cell Sci.* 131. doi:10.1242/jcs.186254.

Goetz, J.G., S. Minguet, I. Navarro-Lérida, J.J. Lazcano, R. Samaniego, E. Calvo, M. Tello, T. Osteso-Ibáñez, T. Pellinen, A. Echarri, A. Cerezo, A.J.P. Klein-Szanto, R. Garcia, P.J. Keely, P. Sánchez-Mateos, E. Cukierman, and M.A. Del Pozo. 2011. Biomechanical remodeling of the microenvironment by stromal caveolin-1 favors tumor invasion and metastasis. *Cell.* 146:148–163. doi:10.1016/j.cell.2011.05.040.

Heissler, S.M., and D.J. Manstein. 2013. Nonmuscle myosin-2: mix and match. *Cell. Mol. Life Sci. CMLS.* 70:1–21. doi:10.1007/s00018-012-1002-9.

7-In summary, the authors have convincingly shown that caveolae play a role in the interaction of melanocytes and keratinocytes in skin pigmentation. Most impressively, caveolae are clearly shown to enhance pigment transfer from melanocytes to keratinocytes (Fig. 4). Since pigment transfer occurs from the end of melanocyte protrusions, it is reasonable that caveolae act at least in part by stimulating protrusion formation and melanocyte-keratinocyte contact, as shown in Fig. 3. The significance of the ability of caveolae to inhibit cAMP formation is less clear. Although in-depth mechanistic characterization of this process remains for the future, the observations presented here represent an intriguing first step.

We thank the reviewer for his/her positive comment. Even if we did not further explore how caveolae could restrict the production of cAMP, the lower increase in cAMP upon treatment of stimulated melanocytes after treatment with the Cav1 CSD mimicking peptide tends to point for a direct role of Cav1 in such regulation. However, and following the recommendation of the referee#3, we have further characterized the consequences on pigmentation following the increase of the cAMP level in stimulated Cav1-depleted melanocytes.

In melanocytes, the molecular events linking cAMP to an elevation of the pigmentation are known to be due to the activation of PKA that will in turn activate (by phosphorylation) the CREB transcription factor (p-CREB). p-CREB activates the transcription of MITF, another transcription factor that is a key regulator of the pigmentation. MITF binds to specific sequences contained in the promoters of melanogenic genes and activates their transcription, thereby increasing their expression (e.g. TYR, DCT, Rab27a) and the melanocyte pigmentation (Buscà and Ballotti, 2000).

We performed additional experiments to better connect the increase of cAMP that we observe in Cav1-depleted melanocytes to the concomitant increase in melanin production, Tyrosinase and DCT expression levels and number of pigmented melanosomes (new **Fig.2** and **Supplementary Fig.2**).

We assessed the phosphorylation of CREB (p-CREB) by immunoblot as a read-out of the activation of PKA by cAMP. Similarly to control cells, whole lysates of Cav1-depleted melanocytes grown 15 min in supplemented medium or in poor medium + forskolin showed increased levels of p-CREB (new **Supplementary Fig.2d**). This indicates that the cAMP produced in stimulated and Cav1-depleted melanocytes can lead to CREB activation (by phosphorylation). Given that CREB was activated (p-CREB), we measured by q-PCR the mRNA expression levels of specific MITF target genes (i.e. TYR, DCT and Rab27a). As compared to control, Cav1-depleted melanocytes stimulated for 3h in the supplemented

medium or in poor medium + FSK showed increased expression levels of TYR and DCT mRNAs, but not of Rab27a mRNA (new **Fig.2c** and **Supplementary Fig.2e**). These results are consistent with the increased protein expression levels of TYR (**Fig.2d**) and DCT (**Supplementary Fig.2f**), and with the stable expression levels of Rab27a observed in Cav1-depleted melanocytes (**Supplementary Fig.2h**). Since TYR and DCT are two melanin-synthesizing enzymes, these results are also consistent with the increased production of intracellular melanin (**Fig.2e**) and the higher number of Stage IV pigmented melanosomes quantified in Cav1-depleted melanocytes (**Fig.2f-g** and **Supplementary Table 2**).

The new data **reinforce the connection between caveolae, the control of cAMP production and the pigmentation of melanocytes through the activation of the transcription of target pigmentation genes**. Notes have been added **page 3 (153)- p8-9 (1199-209)- p9 (1223)**.

Buscà, R., and R. Ballotti. 2000. Cyclic AMP a key messenger in the regulation of skin pigmentation. Pigment Cell Res. 13:60–69. doi:10.1034/j.1600-0749.2000.130203.x.

We would like to thank again the reviewer for the very detailed and constructive insights about our submitted manuscript.

Reviewer #2 (Remarks to the Author):

In this manuscript the authors utilized a wide range of molecular and cell biology working models and modern technologies to characterize the role of caveolae in skin pigmentation. They demonstrate caveolae polarize in melanocytes and are particularly abundant at melanocyte-keratinocyte interface; depletion caveolae by using RNAi in melanocytes increased intracellular cAMP level and melanin pigments, but decreases cell protrusions, cell-cell contacts, pigment transfer and epidermis pigmentation. In addition, they also show caveolae in melanocytes can be regulated by ultra-violet radiations and miR-203a secreted from keratinocytes. These data first-time establish caveolae as a key mediator for the physiological regulations between melanocytes and keratinocytes. This is a novel study with comprehensive data sets that strongly support the conclusions. It will help the field to establish the physiological roles caveolae in skin biology.

We would like to thank the reviewer for the very enthusiastic comment on the submitted study.

Major question: The roles of caveolae in melanocytes are broadly related to signal transduction (cAMP), cell-cell contact, and mechanical sensing. It looks like three all play significant roles in melanocytes functions and pigmentation, which are well supported by the data. However, some of them are not convincing. For example, in cAMP signaling, the authors state that "... the production of cAMP in melanocytes through direct binding to the Cav1-CSD." Although Cav1 CSD mimics peptide has effect, it doesn't mean this is through a native cav1 CSD direct binding mechanism. Considering previous report (Dev Cell. 2012, 17;23(1):11), other possibilities may exist, and this need to be further discussed.

The reviewer raises an important point. Indeed, we did not demonstrate that the Cav1 CSD motif binds directly to signaling molecules in order to affect their activities. We cannot exclude that the Cav1-CSD peptide impacts the production of cAMP by other mechanisms still related to Cav1, but independent on its direct interaction with some partners (e.g. transmembrane Adenylyl Cyclase).

The report mentioned by the reviewer (Collins et al., 2012) brings strong arguments against a direct interaction of the Cav1 CSD with Caveolin-binding motif (CBM) containing proteins that are present in a plethora of putative Cav1- or caveolae-associated interactors. Indeed, the molecular mechanism underlying the "activity" of the Cav1 CSD peptide remains poorly characterized. However, several different studies have successfully used the Cav1 CSD and have reported its potent efficiency in controlling various cellular processes related to Cav1 and caveolae biology. For instance, CSD peptide can interact with endothelial Nitric Oxide (NO) synthase and was shown to decrease NO production and release by endothelial cells. This causing likely a lower inflammation in *in vivo* models (Bernatchez et al., 2005). Also, cells treated with CSD peptide displayed stabilized focal adhesion structures through a decrease of the turnover of some of their core components, leading to a reduced cell migration and motility (Meng et al., 2017). In addition, CSD peptide has been shown to reduce the cAMP production via the control of Adenylyl Cyclase activity (Toya et al., 1998). **Therefore and because the mode of action of the CSD is still debated (Collins et al., 2012), we propose to not refer to the multiple hypotheses that would explain the observed result presented in our submitted manuscript. However, we have amended in the revised manuscript our conclusions** within the data and discussion sections in order to highlight that other possibilities than the direct interaction can exist:

L189, page 8: "Several studies have reported that caveolae could regulate the activity of various signaling molecules, mostly in an inhibitory fashion, through direct binding to the caveolin-1 scaffolding domain (CSD;^{61,62})" **has been modified by** "Several studies have reported that caveolae could regulate the activity of various signaling molecules, mostly in an inhibitory fashion **and in a process dependent of the caveolin-1 scaffolding domain**^{40,41} (CSD)."

L195, page 8: “These results strongly suggest that caveolin-1 reduces the activity of tmACs and the production of cAMP in melanocytes through direct binding to the Cav1-CSD.” **has been modified by** “These results strongly suggest that caveolin-1 through its CSD leads to the decrease of the production of cAMP in melanocytes, possibly by reducing the activity of tmACs.”

L374, page 14: “This shows that caveolae mitigate the cAMP-dependent signaling in melanocytes, likely through Cav1 binding to tmACs and direct inhibition of their catalytic activity.” **has been modified by** “So, and even if the mode of action of the CSD is still debated⁵⁸, this shows that caveolae mitigate the cAMP-dependent signaling in melanocytes through the possible interaction of Cav1 with tmACs to inhibit their catalytic activity.”

Bernatchez, P.N., P.M. Bauer, J. Yu, J.S. Prendergast, P. He, and W.C. Sessa. 2005. Dissecting the molecular control of endothelial NO synthase by caveolin-1 using cell-permeable peptides. *Proc. Natl. Acad. Sci. U. S. A.* 102:761–766. doi:10.1073/pnas.0407224102.

Collins, B.M., M.J. Davis, J.F. Hancock, and R.G. Parton. 2012. Structure-based reassessment of the caveolin signaling model: do caveolae regulate signaling through caveolin-protein interactions? *Dev. Cell.* 23:11–20. doi:10.1016/j.devcel.2012.06.012.

Meng, F., S. Saxena, Y. Liu, B. Joshi, T.H. Wong, J. Shankar, L.J. Foster, P. Bernatchez, and I.R. Nabi. 2017. The phospho-caveolin-1 scaffolding domain dampens force fluctuations in focal adhesions and promotes cancer cell migration. *Mol. Biol. Cell.* 28:2190–2201. doi:10.1091/mbc.E17-05-0278.

Toya, Y., C. Schwencke, J. Couet, M.P. Lisanti, and Y. Ishikawa. 1998. Inhibition of adenylyl cyclase by caveolin peptides. *Endocrinology.* 139:2025–2031. doi:10.1210/endo.139.4.5957.

Minor: Fig 1i, it states “... n=2 independent experiments”. How is the p-value or SE calculated?

To increase the statistical significance, we have performed additional experiments (now n=3). The original conclusion is unchanged by showing that Cav1 protein expression level in melanocytes is statistically increased after 3 days of consecutive stimulation with solar-mimicking UV-B radiations (see new **Fig.1i**).

Reviewer #3 (Remarks to the Author):

In this manuscript, Domingues et al. investigate the role of caveolae in melanocyte-keratinocyte communication and pigmentation. The authors observed an accumulation of caveolae proteins in melanocytes cocultured with keratinocytes. They depleted core structural components of caveolae in primary melanocytes and reported altered melanocyte morphology and hyperpigmentation compared to control cells. Furthermore, Cav1-depleted melanocytes were more susceptible to bursting upon hypo-osmotic shock, suggesting an altered membrane integrity when caveolae are dysfunctional. The data presented in this study are interesting and provide evidence of caveolae as part of the skin pigmentation process. However, several findings are over interpreted and currently not sufficiently connected, and the manuscript title is not supported by the data presented. The findings on Cav1 function in cAMP signaling and melanocyte morphology do not strengthen the authors' claims, as neither have been linked to skin pigmentation. It remains open, in which step of skin pigmentation (sensing of pigmentation inducing signals, melanin production, melanosome packaging, melanosome transport, dendrite development, melanocyte-keratinocyte contact, melanin transfer) caveolae are important.

We would like to thank the reviewer for finding our study of interest and for raising constructive criticisms.

We have provided additional data that better connect the Cav1 function in cAMP signaling, melanocyte pigmentation and morphology in response to signals from keratinocytes. Altogether, our results strengthen our original findings that caveolae are required for the physiological response of melanocytes to cues from keratinocytes (e.g. modulation of pigmentation inducing signals, melanin production, dendrites formation, melanocyte-keratinocyte contacts and melanin transfer). We have also provided additional data to better evidence the caveolae-dependent coupling of signaling and mechanics in melanocytes by (see point#5 of the same reviewer), therefore, we would like to keep the original title.

Major points:

1. The accumulation of Cav1 and Cavin in melanocytes cocultured with keratinocytes is an interesting finding. How do caveolae proteins distribute in melanocytes equally surrounded by keratinocytes? What do the authors mean with the term "polarized" in this context?

We apologize for not being clear in the initial manuscript. We do not believe that caveolae (in 2D) must be "polarized" at the plasma membrane area of the melanocyte in apposition to keratinocytes. The confusion might arise from the generic term "polarized" (and we agree that it was misused) as well as from the data obtained from 3D reconstructed epidermis.

First, the reconstructed epidermis is made of normal melanocytes and keratinocytes seeded on top of a dead dermis (see Methods section) and cultured at the air/liquid interface. Therefore, and due to the low number of melanocytes (alike to the proportion described in the normal human epidermis; approximately 1 melanocyte to 40 keratinocytes), the plasma membrane of a melanocyte is almost exclusively facing the one of a keratinocyte. Given that our quantitative analysis was made on the reconstructed epidermis (and qualitatively validated in human skin biopsies), we only focused on the cell-cell interfaces observed in the synthetic tissue. Regarding the melanocyte, it corresponds to the melanocyte-keratinocyte interface. Therefore, the electron microscopy analysis of the reconstructed epidermis identified almost exclusively caveolae at plasma membrane area of the melanocyte found in close apposition to the one of the keratinocyte.

Second, we have exploited a 2D co-culture system. In that context and, as stated by the reviewer, melanocytes are "*equally surrounded by keratinocytes*". Our first result (**Fig.1c**, light grey bars) showed that one third of the population of melanocytes grown alone displayed a non-homogenous distribution of caveolae (based on Cav1 and cavin1 staining). We would like to stress here that melanocytes (even if not in co-culture) have thus the intrinsic capacity to

distribute caveolae asymmetrically at their plasma membrane (see **Supplementary Fig.1a**). Interestingly, the proportion of co-cultured melanocytes that displayed an asymmetrical distribution of caveolae was significantly increased (x2) (**Fig.1c**, white bars). This effect is specific of keratinocytes because melanocytes co-cultured with the irrelevant HeLa cells behaved as grown alone (**Fig.1c**, dark grey bars). That indicates that keratinocytes and/or factors secreted by keratinocytes enhance the autonomous capacity of melanocytes to form and/or stabilize caveolae at the plasma membrane. We have now added new data that further show (**Fig.1d**) that the keratinocytes-conditioned medium (Ker-CM) is sufficient to increase the number of melanocytes presenting an asymmetrical distribution of caveolae. This new data (**Fig.1d**, white bars) show first that secreted factors by keratinocytes can enhance the asymmetrical distribution of caveolae in melanocytes and second that **melanocytes do not need to be in close apposition with keratinocytes in order to distribute asymmetrically their caveolae** (see pages 5-6, I110-120).

In conclusion, our interpretation is that **caveolae in melanocytes do not distribute to the plasma membrane that must face keratinocytes, but instead to some plasma membrane domains of melanocytes that would be compatible with caveolae biogenesis and/or with their redistribution/ stabilization** (see page 14, I360).

Therefore, we have replaced the term “polarized” at numerous places within the text by **“asymmetrically distributed”** or related terms.

It seems surprising that caveolae do not accumulate at the tips of the dendrites, where melanin transfer mainly occurs but rather at perinuclear sides and at the lateral sides.

Melanin transfer has been proposed to occur mainly at the tip of the dendrites of melanocytes. However, to our knowledge, a formal proof is still lacking (and especially in tissue). Therefore, the mode of transfer of melanin is still debated (see the review by Wu and Hammer, (Wu and Hammer, 2014)). For instance, the peripheral positioning of pigmented melanosome (and especially at the tip of dendrites) requires at least the small GTPase Rab27a (see Fig.2 in (Wu et al., 2001)). However, surprisingly, Rab27a-depleted melanocytes did not show any defect in secreting pigment in the presence of keratinocytes (see Fig.4 in (Tarafder et al., 2013)), suggesting that melanin secretion is not solely occurring at dendrite tips, but also at other areas of the plasma membrane.

It is correct that we did not observe a robust Cav1 or cavin1 staining at the tip of dendrites of melanocytes in 2D culture. While not excluding that few caveolae could be there, they are at least not abundantly present.

We would like to clarify our view for the referee. We do not believe that caveolae-positive plasma membrane domains correspond to the preferential area supporting the melanin transfer. Our results suggest instead that the asymmetric distribution of caveolae could regionalize the intracellular signaling in the melanocyte and its capacity to remodel its plasma membrane and to contact the keratinocyte in order to ultimately support the melanin transfer. Therefore, we anticipate that **not having caveolae at the tip of the dendrites is fully compatible with the capacity of melanocyte to secrete pigment at those very distal regions.**

Tarafder, A.K., G. Bolasco, M.S. Correia, F.J. Pereira, L. Iannone, A.N. Hume, N. Kirkpatrick, M. Picardo, M.R. Torrisi, I.P. Rodrigues, J.S. Ramalho, C.E. Futter, D.C. Barral, and M.C. Seabra. 2013. Rab11b Mediates Melanin Transfer between Donor Melanocytes and Acceptor Keratinocytes via Coupled Exo/Endocytosis. *J. Invest. Dermatol.* doi:10.1038/jid.2013.432.

Wu, X., and J.A. Hammer. 2014. Melanosome transfer: it is best to give and receive. *Curr. Opin. Cell Biol.* 29:1–7. doi:10.1016/j.ceb.2014.02.003.

Wu, X., K. Rao, M.B. Bowers, N.G. Copeland, N.A. Jenkins, and J.A. Hammer 3rd. 2001. Rab27a enables myosin Va-dependent melanosome capture by recruiting the myosin to the organelle. *J Cell Sci.* 114:1091–100.

2. From conditioned-media experiments, the authors conclude that asymmetric caveolae distribution is caused by keratinocyte-secreted factors. Mechanistically, such result would be very interesting. The methods section, however, suggests that with the indicated melanocyte medium an inappropriate control was used. The keratinocyte-conditioned medium contains keratinocyte medium supplements (DermaLife K Life), whereas the melanocyte medium contains Dermalife M life supplements. Thus any differences observed in melanocytes incubated with either melanocyte medium or keratinocyte-conditioned (keratinocyte) medium are not necessarily caused by secreted keratinocyte-derived factors but could also be due to the difference in the commercial media supplements. The authors should use the same medium for both cell types or provide essential negative controls to justify their claim.

We agree with the reviewer that the appropriate negative control consists of melanocytes grown in keratinocyte medium only (DermaLife Basal Medium supplemented with DermaLife K Life). Because normal human melanocytes do not tolerate long-term culture in such medium without factors secreted by keratinocytes (e.g. co-culture or culture in keratinocyte-conditioned medium) or factors externally added (our unpublished observations), we performed the requested experiment by examining melanocytes after 10h of culture in keratinocyte medium (Ker medium; new **Fig.1d**).

Melanocytes grown on coverslips were fixed and further processed by immunofluorescence using anti-Cav1 or anti-Cavin1 antibody. We then quantified the number of melanocytes displaying an asymmetrical distribution of caveolae staining. The new data (**Fig.1d**, dark grey bars) show that the number of melanocytes with an asymmetrical distribution of caveolae was not potentiated when grown alone in keratinocyte medium, and as compared to melanocytes grown in keratinocyte-conditioned medium. **This data reinforces the original finding showing that the factors secreted by keratinocytes potentiate the asymmetric distribution of caveolae by the melanocyte.** We have added a note referring to this new data in **pages 5-6 (I112-120) and 14 (I358-361)**.

3. The authors provide evidence that Cav1 controls cAMP signaling, and propose that this is an upstream event of controlling pigmentation in their system. However, cAMP is involved in many signaling pathways, not only in pigmentation, and no data are provided that altered cAMP levels indeed cause altered melanin synthesis in their system. This part requires further mechanistic exploration to test whether Cav1 controls pigmentation through cAMP regulation.

We performed additional experiments to better connect the increase of cAMP that we have observed in Cav1-depleted melanocytes to the concomitant increase in melanin production, Tyrosinase and DCT expression levels, and number of pigmented melanosomes (new **Fig.2** and **Supplementary Fig.2**).

In melanocytes, the molecular events linking cAMP to an elevation of the pigmentation are known to be due to the activation of PKA that will in turn activate (by phosphorylation) the CREB transcription factor (p-CREB). p-CREB activates the transcription of MITF, another transcription factor that is a key regulator of the pigmentation. MITF binds to specific sequences contained in the promoters of melanogenic genes and activates their transcriptions, thereby increasing their expression (e.g. TYR, DCT, Rab27a) and the melanocyte pigmentation (Buscà and Ballotti, 2000).

We assessed the phosphorylation of CREB (p-CREB) by immunoblot as a read-out of the activation of PKA by cAMP. Similarly to control cells, whole lysates of Cav1-depleted melanocytes grown 15 min in supplemented medium or in poor medium + forskolin showed increased levels of p-CREB (new **Supplementary Fig.2d**). This indicates that the cAMP produced in stimulated and Cav1-depleted melanocytes can lead to CREB activation (by phosphorylation). Given that CREB was activated (p-CREB), we measured by q-PCR the mRNA expression levels of specific MITF target genes (i.e. TYR, DCT and Rab27a). As compared to control, Cav1-depleted melanocytes stimulated for 3h in the supplemented medium or in poor medium + FSK showed increased expression levels of TYR and DCT

mRNAs, but not of Rab27a mRNA (new **Fig.2c** and **Supplementary Fig.2e**). These results are consistent with the increased protein expression levels of TYR (**Fig.2d**) and DCT (**Supplementary Fig.2f**), and with the stable expression levels of Rab27a observed in Cav1-depleted melanocytes (**Supplementary Fig.2h**). Since TYR and DCT are two melanin-synthesizing enzymes, these results are also consistent with the increased production of intracellular melanin (**Fig.2e**) and the higher number of Stage IV pigmented melanosomes quantified in Cav1-depleted melanocytes (**Fig.2f-g** and **Supplementary Table 2**).

The new data **reinforce the connection between caveolae, the control of cAMP production and the pigmentation of melanocytes through the activation of the transcription of target pigmentation genes**. Notes have been added **page 3 (I53)- p8-9 (I199-209)- p9 (I223)**.

Buscà, R., and R. Ballotti. 2000. Cyclic AMP a key messenger in the regulation of skin pigmentation. Pigment Cell Res. 13:60–69. doi:10.1034/j.1600-0749.2000.130203.x.

4. A key finding of this manuscript is the involvement of caveolae proteins in pigmentation. Late stage melanosomes and melanin synthesis proteins like tyrosinase are enriched upon loss of Cav1. The authors claim that caveolae control pigmentation via cAMP, which as stated has not been demonstrated. Instead, various other data in this manuscript together suggest that the primary defect of Cav1 depletion is an inability to release pigment (e.g. more stage IV melanosomes, less pigment transferred to keratinocytes). Thus it seems possible that Cav1 promotes the transfer rather than the initial synthesis of melanin.

The new included data in response to the point 3 (see above) argue that Cav1 controls the initial step of melanin synthesis by the control of the cAMP production and the subsequent activation of the transcription of the genes coding for melanin-synthesizing enzymes.

To dissect the mechanism of Cav1-mediated pigment regulation the authors could determine if transcription of melanin synthesis genes is altered by forskolin treatment and Cav1 depletion. If Cav1 indeed acts through cAMP one would expect this. If, however, the Tyr and DCT increase is only detectable on protein level, this might argue for an indirect effect caused by accumulation of late stage melanosomes within cells and hence associated protein machineries. This could perhaps also explain why the authors did not detect differences in Rab27a levels.

We performed the experiment suggested by the reviewer as detailed on the point 3 (see above). As compared to control, the new analysis shows that Cav1-depleted melanocytes incubated 3h in Mel-supplemented medium or poor medium supplemented with FSK (30uM) display an increased expression levels of TYR and DCT mRNAs, but not of Rab27a mRNAs. The new results are shown in **Fig.2** and **Supplementary Fig.S2**. Together with the aforementioned data, it shows that **Cav1 modulates the initiation of melanocyte pigmentation via cAMP-dependent signaling pathways**. It also indicates that the increased pigmentation observed in Cav1-depleted cells is likely due to **both an increased neo-production of melanin and to a reduced melanin secretion**. Thus, it re-emphasizes the double function of Cav1 during pigmentation by regulating melanin synthesis and its transfer as mentioned on **page 15 (I399)** of the revised manuscript.

5. The title and abstract suggest that Caveolae control melanocyte mechanics. This, however, has not been demonstrated, hence the terminology used throughout the manuscript is misleading. In figure 3a-e, morphological changes in Cav1-depleted melanocytes are presented. The authors call this consistently “mechanical behavior” although no mechanical parameters have been assessed. Though the hypo-osmotic shock assay is closer to probing potential differences in membrane tension, it also bears limitations given the lack of knowledge how siRNA-mediated depletion of Cav1 affects the overall viability. In fig. 3f, t=0, siCav1 cells seem more round and possibly more PI-positive, but no quantitative data are provided for this time point. A careful characterization of proliferation and apoptosis following Cav1 depletion is

required, as a higher susceptibility to cell death could result in a less protrusion-rich morphology and higher number of PI-positive cells, which the authors now attribute to mechanical defects. Similarly, the altered morphology might be a consequence of altered cell-matrix adhesion. Independent of above limitations, the current morphological data have not been properly connected to the main question: how does Cav1 promote skin pigmentation? One may consider leaving this whole set out.

The reviewer has raised an important point that we have addressed (i) by examining the proliferation and cell death of Cav1-depleted melanocytes, and (ii) by providing additional molecular insights on how Cav1 controls changes of cell morphology via the regulation of the activity of the non-muscle myosins-2 motor (NMMIIs).

(i) First, we have exploited the experimental set-up described in the new **Fig.3h** to examine the number of melanocytes (depleted or not for Cav1) that were propidium iodide (PI)-positive before being subjected to the hypo-osmotic shock (i.e. at T=0min). The PI dye can only enter the cell and intercalate the DNA if plasma membrane integrity is lost. Given that the PI dye is not permeant to live and intact cells, it can be also used to label and quantify the number of dead cells, including those subjected to apoptosis (Baskić et al., 2006). In control and Cav1-depleted melanocytes, we did not observe any statistical differences in the number of PI-positive cells (**Fig. 1a for reviewer#3**; n=3 experiments), showing that Cav1-depletion in melanocytes did not result in increased cell death, including apoptosis. Then, we examined the percentage of round cells in the hypo-osmotic experiment to test whether the depletion of Cav1 could globally affect the adhesion of melanocytes. We did not observe any statistical positive correlation (**Fig. 1b for reviewer#3**; n=3 experiments) between the number of round melanocytes in control- and Cav1-depleted condition. Finally, we have examined the cell proliferation by counting the number of melanocytes before and after 3 days of Cav1 inactivation by siRNA. We did not observe any statistical differences in the proliferation of Cav1-depleted melanocytes as compared to control-depleted cells (**Fig. 1c for reviewer#3**; n=3 experiments). Note that because normal melanocytes originate from different donors, their proliferation rate varies in between experiments. So, together with data presented in Fig1b for reviewer#3, that suggests that the adhesion and number of cells still attached to the substrate are similar in control and Cav1-depleted cells.

Fig.1 for reviewer#3 – Quantification of PI-positive melanocytes at T=0 (a), of round cells (b) and of the cell number ratio before and after siRNA treatment (c). Related to Fig. 3h,i of the submitted manuscript.

In conclusion, **Cav1 depletion in melanocytes does affect neither the cell proliferation nor cell death**. We have included these data to the attention of the reviewer (see figures above). If explicitly requested, we will include these data in the revised manuscript as supplemental data.

Baskić, D., S. Popović, P. Ristić, and N.N. Arsenijević. 2006. Analysis of cycloheximide-induced apoptosis in human leukocytes: fluorescence microscopy using annexin V/propidium iodide versus acridin orange/ethidium bromide. *Cell Biol. Int.* 30:924–932. doi:10.1016/j.cellbi.2006.06.016.

(ii) To better address whether Cav1 controls the morphology of melanocyte by affecting its cell mechanics, we have investigated whether the activity of the non-muscle myosins-II (NMMIIs) was required for melanocytes morphology. The rationale is the following. Changes in cell shape are mainly driven by the plasma membrane-associated subcortical actomyosin network. There, myosins and especially NMMIIs generate contractile forces that create a tension translated into plasma membrane deformations (see a recent review from Ewa Paluch's lab, (Chugh and Paluch, 2018)). The cortical tension can be thus modulated by the activity of myosins-II that is largely associated with the phosphorylation status of the regulatory Light Chain (MLC). Therefore, many cellular studies report the phosphorylation of MLC (p-MLC) as an indicator of NMMIIs activity and thus as a read-out of the contractile force generated by NMMIIs (Heissler and Manstein, 2013).

Cav1-depleted melanocytes stimulated by forskolin or with the supplemented medium showed less MLC phosphorylation (p-MLC) as compared to control (see new **Fig.3f-g**). So, it suggests that **less contractile forces were generated in absence of caveolae, leading to a lower cortical tension of the plasma membrane of the Cav1-depleted melanocytes**. It has to be noted that such influence of Cav1 expression on p-MLC is not restricted to melanocytes because it was already reported in Cav1-KO fibroblasts (Goetz et al., 2011).

In conclusion, Cav1-depleted melanocytes display lower NMMIIs activity in response to increased cAMP production (see new **Fig.3f-g**). This would result in diminished contractile forces and, consequently, less cortical tension generated on the plasma membrane. Together with the hypo-osmotic shock experiment (**Fig.3h-i**), it suggests that melanocytes devoid of caveolae displayed a lower contractility as compared to control cells. Given that a proper adjustment of the cell contractility is most likely necessary for changes in cell morphology, this may explain the dramatic failure of melanocytes to extend dendritic-like protrusions in the absence of caveolae (**Fig. 3a-b**).

Taken together, the new data suggests that **Cav1 and caveolae control the cAMP-dependent changes in shape and dendricity of melanocytes by modulating the contractile force generated by the actomyosin subcortical network. Such force is likely relying on the activity of the NMMIIs and is needed to protrude the plasma membrane of melanocyte in response to signaling.**

We have described the new data (**pages 11-12, I285-294 and page 12, I311**) and further discussed them (**page 15, I378; page 16, I418**). We believe that the additional data included in the revised manuscript strongly reinforce the role of caveolae in melanocytes mechanics and better couple this mechanical aspect to the control of signaling by caveolae in melanocytes during pigmentation.

Chugh, P., and E.K. Paluch. 2018. The actin cortex at a glance. *J. Cell Sci.* 131. doi:10.1242/jcs.186254.

Goetz, J.G., S. Minguet, I. Navarro-Lérida, J.J. Lazcano, R. Samaniego, E. Calvo, M. Tello, T. Osteso-Ibáñez, T. Pellinen, A. Echarri, A. Cerezo, A.J.P. Klein-Szanto, R. Garcia, P.J. Keely, P. Sánchez-Mateos, E. Cukierman, and M.A. Del Pozo. 2011. Biomechanical remodeling of the microenvironment by stromal caveolin-1 favors tumor invasion and metastasis. *Cell.* 146:148–163. doi:10.1016/j.cell.2011.05.040.

Heissler, S.M., and D.J. Manstein. 2013. Nonmuscle myosin-2: mix and match. *Cell. Mol. Life Sci. CMLS.* 70:1–21. doi:10.1007/s00018-012-1002-9.

6. The authors used the 3D-HRPE model and observed reduced melanin transfer in Cav1-depleted melanocytes. However, an overall reduced number of melanocytes could as well explain reduced melanin content in keratinocytes during the establishment of the 3D-HRPE model. The depigmented spot in the middle of siCav1-HRPE seems surprising but was not explained. It might be a hint for lower numbers of melanocytes. The number of melanocytes

should be examined e.g. in HRPE cross-sections, and the nature of the pigmentation pattern should at least be discussed. Moreover, quantitative data are required.

We believe that the lighter pigmentation of the siCav1-HRPE reflected well the lower melanin transfer observed in the tissue. It is however true that it could be due to a smaller number of melanocytes present in the epidermis. Given that we did not observe increased cell death *in vitro* (see above point#5 to the same referee), we do not expect that this would be the case in tissue. To reinforce our assumption, we counted the number of melanocytes and keratinocytes observed on the EM micrographs of the resin embedded 3D-HRPE. This set of micrographs was randomly acquired across different ultrathin cross-sections and are the same that were used to quantify the number of pigment granules per keratinocyte (**Fig.4e**).

In control HRPEs, we counted 17 melanocytes and 61 keratinocytes, corresponding to a Mel:Ker ratio of 0,28 (1:3,6).

In Cav1-depleted HRPE, we counted 13 melanocytes and 35 keratinocytes, corresponding to a Mel:Ker ratio of 0,37 (1:2,7). These data have been included in the method section (**page 19, I525**).

The Mel:Ker ratios are in the same range for the two conditions and show even an increased proportion of Cav1-depleted melanocytes per keratinocyte when compared to control HRPE. That would support that no particular cell death is associated with Cav1-depleted melanocytes in the tissue as we have now described *in vitro* (see above point#5). That leads us to exclude that the lower pigmentation of siCav1-HRPE is due to a dramatic loss of the Cav1-depleted melanocytes.

One key criterion regulating skin color is the amount of pigments found in keratinocytes. We have added a note **page 16 (I408)** to explicitly refer to that fact. Keratinocytes are much more abundant than melanocytes in tissue and most of the pigment in skin resides in keratinocytes (Hurbain et al., 2017). Another evidence comes from the coat color of mice KO for factors that regulate melanin transfer (e.g. ashen mice KO for Rab27a). While Rab27a KO melanocytes are heavily pigmented, the peripheral distribution of the pigment produced in melanosomes is dramatically affected as melanosomes accumulate in the perinuclear area (Hume et al., 2001). The acquisition of the pigment by keratinocytes is thus significantly decreased and results in coat color dilution in ashen mice (Wilson et al., 2000) and in an albinism-like phenotype in patients suffering from the Griscelli Syndrome (Bowman et al., 2019). Therefore, pigment in keratinocyte is crucial for skin/ hair color.

Our *in vitro* observation showed that the number of keratinocytes positive for pigment is lower when co-cultured with siCav1-depleted melanocytes as compared to control (**Fig.4b**). Moreover, when keratinocytes are positive for pigment, they contain less pigment than the ones found in keratinocytes from the control co-culture (**Fig.4c**). This result is recapitulated in Mel siCav1 HRPE as compared to control tissue (**Fig.4e**). However and as expected from our *in vitro* data (**Fig.2**), siCav1-depleted melanocytes observed in the reconstructed tissue are still heavily pigmented (**Fig.4d**). Therefore, the lighter coloration of Mel siCav1-HRPE (as compared to the Mel si-Ctrl-HRPE) tissue is most likely due to the low amount of pigment found in keratinocytes. Together, the most reasonable explanation is that **a lower transfer efficiency of pigment from siCav1-depleted melanocytes to keratinocytes is responsible of the lowest amount of pigment found in keratinocytes that ultimately lead to a less color complexion of the tissue.**

Bowman, S.L., J. Bi-Karchin, L. Le, and M.S. Marks. 2019. The road to lysosome-related organelles: Insights from Hermansky-Pudlak syndrome and other rare diseases. *Traffic Cph. Den.* 20:404–435. doi:10.1111/tra.12646.

Hume, A.N., L.M. Collinson, A. Rapak, A.Q. Gomes, C.R. Hopkins, and M.C. Seabra. 2001. Rab27a regulates the peripheral distribution of melanosomes in melanocytes. *J Cell Biol.* 152:795-808.

Hurbain, I., M. Romao, P. Sextius, E. Bourreau, C. Marchal, F. Bernerd, C. Duval, and G. Raposo. 2017. Melanosome distribution in keratinocytes in different skin types: melanosome clusters are not degradative organelles. *J. Invest. Dermatol.* doi:10.1016/j.jid.2017.09.039.

Minor points:

1. The manuscript text would benefit from proof-reading. Often plural forms were used where singular would be preferred. For instance, the title should read: “Caveolae coupling of melanocyte signaling and mechanics is required for...”, or “keratinocyte-secreted”.

We apologize. The revised manuscript has been now proofread by a native English.

2. In many blots, the authors show bar diagrams, representing only the mean and SEM. All single data points should be displayed. Moreover, number of replicates should be stated in each legend, unambiguously for each data set.

We have now provided the information requested by the referee.

3. The gene PMEL encodes a melanocyte-specific type 1 transmembrane glycoprotein and is not equal to melanin. For clarity, the authors should strictly use the name of the antibody/antigen in Fig. 1a, Suppl. Fig.1a and Fig.4a.

We originally referred to “melanin” for the readers outside the pigmentation field, but we are now consistently referring to HMB45 antibody that recognizes a processed form of PMEL. We explicitly refer to that antibody in the method section **page 17, I442**.

4. The data in fig. 2f are displayed as percentage. An absolute increase in stage IV melanosomes would result in relative lower percentage of stage II melanosomes. How do the absolute numbers of melanosomes at different stages depend on Cav1 expression? Is there really a reduction of immature melanosomes?

The prediction of the reviewer is correct. We have added the raw data below to answer the concern raised. The total numbers for stage II melanosomes are slightly decreased in siCav1 when compared to siCtrl melanocytes (147 vs. 194) while the difference observed for stage IV melanosomes is much more striking (285 vs. 609) between siCtrl- and siCav1-depleted melanocytes. Normalization of the data to the total number of melanosomes quantified, reduced by half the number of stage II melanosomes for siCav1 melanocytes (**Fig.2g**) while decreasing the differences for stage IV melanosomes for 12% when compared to control cells. Still, we believe that an interpretation of the data is that siCav1-depleted melanocytes have a tendency to decrease the number of stage II melanosomes while increasing stage IV melanosomes. The total absolute values are now provided in **Supplementary Table 2 (page 38) and this raw data can will be also available in the source data file.**

siCtrl cell	Stage I	Stage II	Stage III	Stage IV	siCav1 cell	Stage I	Stage II	Stage III	Stage IV
1	4	28	68	67	1	0	16	23	15
2	2	18	45	16	2	1	9	45	48
3	4	22	59	25	3	1	18	83	40
4	0	18	39	5	4	0	7	64	37
5	1	13	35	6	5	0	3	25	70
6	5	5	13	49	6	1	4	33	61
7	0	3	12	31	7	3	1	42	57
8	0	8	39	39	8	0	3	34	96
9	0	3	11	26	9	0	0	9	50

10	0	16	30	0	10	0	13	78	12
11	0	6	43	4	11	0	13	103	11
12	0	4	44	1	12	1	6	58	9
13	4	10	39	3	13	0	9	160	16
14	0	3	39	1	14	1	3	66	29
15	0	6	37	3	15	1	9	96	7
16	0	7	60	7	16	0	10	43	13
17	0	0	42	0	17	3	10	67	12
18	0	14	83	2	18	5	3	72	10
19	0	10	49	0	19	2	10	57	16
TOTAL	20	194	787	285	TOTAL	19	147	1158	609

5. The micrographs in Suppl. Fig 1a do not reflect asymmetric distribution of caveolae proteins in the melanocyte monoculture.

In such condition (melanocytes grown alone or together with the irrelevant HeLa cells), only 1/3 of the melanocyte population displayed an asymmetric distribution of caveolae, meaning that the asymmetry is not enhanced. We have changed accordingly that panel in order to show the melanocytes that displayed or not an asymmetrical distribution of caveolae.

6. Why does UV-B treatment suppress the expression of Cav1 at day 1 before it increases at day 2 and day 3? The authors should at least discuss this point.

In order to clarify this particular point, we have performed additional experiments and show now no statistical difference in the expression level of Cav1 at day1 of UV-B treatment (new Fig. 1i).

7. Fig.1e and Suppl. Fig.1d show the same micrograph. This is not considered scientifically sound.

We did it on purpose in order to first show the images with the manual contour of the cells (Fig.1e) to help the reader to visualize the plasma membrane of individual cells, and then to show the same image (but without the manual contour, Supplementary Fig.1d) to allow the reader to judge 'by itself' the plasma membrane deformation that we have pointed. Therefore, we believe that it is important to keep these figures as they are.

8. The y axis descriptions in Fig.2a & 2b should state "cAMP fold change"

9. In line 246-247 in the main text authors write "..., the major axis increased, while the minor axis increased", which should be corrected in "... minor axis decreased"

10. In line 248 in the main text authors refer to Fig.3d, which should be corrected to Fig.3c

11. The micrographs in Fig.4a do not reflect asymmetric Cav1 distribution in the melanocytes.

12. The y axis legend in Fig.4c does not explain to what the fluorescent intensity is normalized to.

We apologize. In fact, normalization stands for values between 0 and 1, which was not the case for this plot. The values are now normalized as expected and a new version of the plot replaces the old one.

The corrections related to minor points 8-12 have been done accordingly.

We would like to thank again the reviewer for the critical evaluation of the submitted manuscript. We hope that the provided answers and additional information now included in the revised manuscript will be satisfying.

REVIEWERS' COMMENTS

Reviewer #2 (Remarks to the Author):

This revised manuscript has satisfactorily address all my questions.

Reviewer #3 (Remarks to the Author):

In this revised manuscript by Domingues et al., the authors have added comprehensive new data important to justify some of the previous claims made. I feel the study adds intriguing new knowledge on the role of caveolae in skin pigmentation.

While the author sufficiently addressed most of my criticism, I still feel that the manuscript title is not adequately supported by the data provided. The one data set on pMLC (new figure 3f,g, immunoblots) in fact suggests higher pMLC in the siCav1 cells in poor medium, thus opposing the effect when stimulated. The authors should add least discuss this point.

Point-by-point answer to the Reviewers

Reviewers' comments:

Reviewer #1 (Editor Remarks to the Author):

As we were unable to contact Reviewer #1 for their report, we, as previously mentioned, asked Reviewer #2 to comment on your responses to Reviewer #1's previous concerns. Reviewer #2 told us in confidence that Reviewer #1's concerns were satisfactorily addressed.

Reviewer #2 (Remarks to the Author):

This revised manuscript has satisfactorily address all my questions.

We are very grateful to the reviewer#2 and would like to sincerely thank her/him. First for having carefully examined and criticized our manuscript, and second to have accepted to review our response to the first reviewer. This is very appreciated and has greatly helped to speed up the revision process.

Reviewer #3 (Remarks to the Author):

In this revised manuscript by Domingues et al., the authors have added comprehensive new data important to justify some of the previous claims made. I feel the study adds intriguing new knowledge on the role of caveolae in skin pigmentation.

While the author sufficiently addressed most of my criticism, I still feel that the manuscript title is not adequately supported by the data provided. The one data set on pMLC (new figure 3f,g, immunoblots) in fact suggests higher pMLC in the siCav1 cells in poor medium, thus opposing the effect when stimulated. The authors should add least discuss this point.

We really the thank the reviewer for her/his positive assessment and would like to discuss the last point that was raised. The new additions to the submitted revised manuscript are marked in blue.

First of all, we would like to summarize the main conclusion of the Fig3f and 3g. The data show that the phosphorylation of MLC (p-MLC) is significantly increased when melanocytes are activated (by supplemented medium or FSK). This increase is no longer observed when caveolin-1 is depleted by siRNA. Therefore, in the absence of caveolae, stimulated melanocytes have decreased NMMIIs activity.

The reviewer is right that p-MLC appeared to be higher in non-stimulated 'resting' siCav1 melanocytes as compared to control cells. But when comparing the p-MLC/MLC ratio of siCTRL vs. siCav1 cells grown in poor medium, no statistical difference was measured. However, the potential higher p-MLC in resting melanocytes depleted of caveolae would be thus indicative of a slight increase of the steady state activation of the NMMIIs (as compared to non-stimulated control melanocytes). Given that the activity of NMMIIs is regulated by p-MLC and results very likely in local changes in cell cortex mechanical properties^{1,2}, such an increase in p-MLC could be predicted to lead to changes of the plasma membrane plasticity and/or tension. That agrees with the central role of caveolae in membrane tension buffering³. However, it has to be noted that, if such plasticity or tension is modulated in resting conditions, it is not accompanied by a modification of the shape of melanocytes; as illustrated by the similar parameters measured in non-stimulated siCTRL- and siCav1-depleted cells (Fig3a,b; FigS3b-d).

Understanding why the absence of caveolae would increase p-MLC level in resting melanocytes without an obvious impact on the cell shape would require many additional

experiments that are beyond the scope of the present study. One could speculate that since Cav1 has been proposed to be connected with actin-filaments (F-actin), its depletion in resting melanocytes might change F-actin dynamics and lead to a slight elevation of the p-MLC. Indeed, in the (Sinha et al. 2011) study⁴ showing the role of caveolae in membrane tension buffering, F-actin dynamics were clearly connected with the caveolae response. More recently, caveolae were shown to respond to substrate stiffness through actin-dependent control of YAP⁵. Additionally, the level of p-MLC might reflect a change in the balance of specific phosphatases and/ or kinases targeting MLC. Whether Cav1 could regulate such balance is to our knowledge currently unknown.

All together, we believe that our data points more generally towards an important role of caveolae and Cav1 on the activity of the NMMIIs that could lead to its differential regulation depending on the activation status of the cell. That Cav1 regulates NMMII activity is to our knowledge unknown and should be of interest to the community. Therefore, we mention now that possibility by adding a note in the discussion section as follows (p15, l388):

“In addition, non-stimulated Cav1-depleted melanocytes seems to display higher p-MLC — yet without impact on the cell shape. That would suggest that caveolin-1 and caveolae might represent key elements modulating the activity of NMMIIs in very different cell types, like melanocytes or fibroblasts⁵⁹, and under resting and stimulated conditions.”

1. Chugh, P. & Paluch, E. K. The actin cortex at a glance. *J. Cell Sci.* **131**, (2018).
2. Heissler, S. M. & Manstein, D. J. Nonmuscle myosin-2: mix and match. *Cell. Mol. Life Sci. CMLS* **70**, 1–21 (2013).
3. Parton, R. G., McMahon, K.-A. & Wu, Y. Caveolae: Formation, dynamics, and function. *Curr. Opin. Cell Biol.* **65**, 8–16 (2020).
4. Sinha, B. et al. Cells respond to mechanical stress by rapid disassembly of caveolae. *Cell* **144**, 402–13 (2011).
5. Moreno-Vicente, R. et al. Caveolin-1 Modulates Mechanotransduction Responses to Substrate Stiffness through Actin-Dependent Control of YAP. *Cell Rep.* **25**, 1622-1635.e6 (2018).